# 🏛 RITUAL: Random Image Transformations as a Universal Anti-hallucination Lever in LVLMs

## Abstract

Recent advancements in Large Vision Language Models (LVLMs) have revolutionized how machines understand and generate textual responses based on visual inputs. Despite their impressive capabilities, they often produce "hallucinatory" outputs that do not accurately reflect the visual information, posing challenges in reliability and trustworthiness. Inspired by test-time augmentation, we propose a simple, training-free method termed RITUAL to enhance robustness against hallucinations in LVLMs. RITUAL introduces random image transformations as complementary inputs during the decoding phase. Importantly, these transformations are not employed during the training of the LVLMs. This straightforward strategy reduces the likelihood of hallucinations by exposing the model to varied visual scenarios, enriching its decision-making process. While transformed images alone may initially degrade performance, we empirically find that strategically combining them with the original images mitigates hallucinations. Specifically, in cases where hallucinations occur with the original image, the transformed images help correct misinterpretations by adjusting the probability distribution. By diversifying the visual input space, RITUAL provides a more robust foundation for generating accurate outputs. Notably, our method works seamlessly with existing contrastive decoding methods and does not require external models or costly self-feedback mechanisms, making it a practical addition. While extremely simple, RITUAL significantly outperforms existing contrastive decoding methods across several object hallucination benchmarks, including POPE, CHAIR, and MME.

## 1 Introduction

Large Vision-Language Models (LVLMs) (Dai et al., 2024; Zhu et al., 2023; Liu et al., 2023c;b; Bai et al., 2023) have emerged as a pivotal technology, enabling machines to interpret complex visual scenes and generate contextually appropriate textual descriptions. These models integrate and process inputs from both visual and linguistic domains, offering unprecedented possibilities in applications ranging from video content creation (Brooks et al., 2024) to assistive technologies (Team et al., 2023; OpenAI, 2023).

Despite their potential, LVLMs are often criticized for generating "hallucinatory" content (Li et al., 2023c; Zhao et al., 2023; Wang et al., 2023b; Huang et al., 2023) – outputs that appear plausible but do not faithfully reflect the visual inputs. This gap in reliability and trustworthiness is particularly concerning for sensitive applications such as medical diagnosis (Zhou et al., 2023a; Liu et al., 2023d), surveillance (Wu et al., 2024; Hasan et al., 2024), and autonomous driving (Li et al., 2024).

The challenge primarily arises from the difficulty in maintaining alignment between the visual inputs and textual outputs, given the complexity of training such models to accurately interpret and narrate visual data. Although several strategies have been developed to mitigate these issues, they often require extensive additional training (Jiang et al., 2023; Zhou et al., 2023b; Gunjal et al., 2023; Liu et al., 2023a; Sun et al., 2023; Wang et al., 2023a; Yin et al., 2023; Lu et al., 2024; Zhai et al., 2024; Yue et al., 2024), sophisticated feedback mechanisms (Yin et al., 2023; Yu et al., 2023; Kim et al., 2024; Sun et al., 2023), or reliance on auxiliary models (Zhao et al., 2024; Wan et al., 2024; Deng et al., 2024; Yang et al., 2024; Li et al., 2023b), which can complicate deployment and scalability.

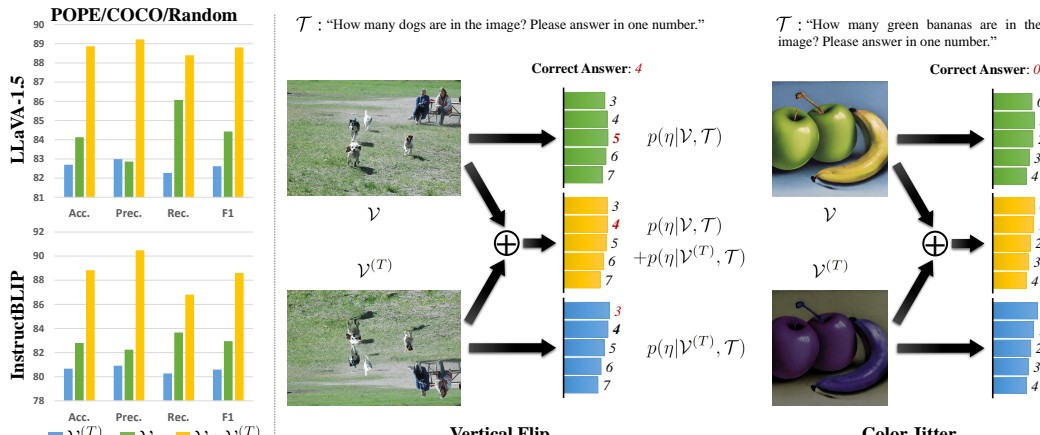

Figure 1: **Intriguing impact of random image transformations on LVLMs. (Left)** Using the randomly transformed image ($\mathcal{V}^{(T)}$) as a visual input to LVLMs (Liu et al., 2023c; Dai et al., 2024) results in lower performance compared to using the original image ($\mathcal{V}$). **(Right)** However, when these two images are used together ($\mathcal{V} + \mathcal{V}^{(T)}$), an intriguing phenomenon is observed: cases incorrectly predicted with the original image are now correctly predicted. (i) Although $\mathcal{V}^{(T)}$ alone does not yield a correct answer, it reduces the likelihood of a hallucinated answer and increases the chances of a correct answer. (ii) In some cases, $\mathcal{V}^{(T)}$ strongly aligns with the correct answer, leading to accurate answers.

We present a simple, training-free approach termed RITUAL, which leverages random image transformations to complement the original image and enhance models' robustness. RITUAL is designed to address the issue of visual hallucination by employing a dual-input strategy that integrates both the original and a randomly transformed image. The final prediction is an ensemble of the individual predictions generated from both the original and augmented images. This provides a more comprehensive visual context, enriching the model's exposure to a diverse array of visual scenarios, thereby enhancing the robustness and reliability of text generation. Much like how humans refine their understanding by observing objects from different angles and under varying conditions, our approach fosters *cognitive flexibility* (Ionescu, 2012) – the ability to adapt to new situations and switch between tasks or concepts.

Our approach builds on the principles of Test-Time Augmentation (TTA) (Zhang et al., 2022; Shanmugam et al., 2021; Pérez et al., 2021), a technique that improves model robustness and generalization at inference time by using multiple augmented versions of an input. TTA is particularly useful in scenarios where the test set exhibits high variance or when inputs contain ambiguities. By generalizing over these uncertainties, TTA helps reduce model sensitivity to minor perturbations, leading to more reliable predictions.

Importantly, these image transformations are applied only during the inference phase, not during training. As demonstrated in Fig. 1 (Left), using transformed images ($\mathcal{V}^{(T)}$) alone initially degrades performance compared to using the original image ($\mathcal{V}$), due to the introduction of novel visual artifacts. However, when the original and transformed images are combined ($\mathcal{V} + \mathcal{V}^{(T)}$) significantly enhances the quality and reliability of the model's outputs. While neither the original image nor the transformed image alone may provide an accurate response, their combination reduces the likelihood of a hallucinated response and increases the chances of a correct answer. In some cases, the transformed image strongly aligns with the correct answer, resulting in accurate predictions.

Our experiments evaluate RITUAL across several benchmarks, including POPE (Rohrbach et al., 2018), CHAIR (Li et al., 2023c), and both MME-Hallucination and MME-Fullset (Fu et al., 2024). Despite its simplicity, RITUAL effectively reduces hallucination across these benchmarks and enhances the general capabilities of LVLMs. Moreover, RITUAL consistently outperforms existing contrastive decoding baselines (Leng et al., 2023; Favero et al., 2024) in all tested benchmarks. RITUAL is also compatible with current contrastive decoding methods, and when used in conjunction, it further amplifies the improvements over these methods.

## 2 RELATED WORK

**Hallucinations in LVLMs.** LVLMs are susceptible to visual hallucinations, in which the generated text descriptions include objects or details entirely irrelevant from the given image. A range of methods has been introduced to address the issue by additional training (Gunjal et al., 2023; Liu et al., 2023a; Sun et al., 2023; Wang et al., 2023a; Yin et al., 2023; Lu et al., 2024; Jiang et al., 2023; Zhou et al., 2023b; Zhai et al., 2024; Yue et al., 2024). While these approaches offer promise, they often face practical limitations due to their dependence on additional data and extensive training periods. In response to these limitations, training-free approaches have gained traction. These models aim to refine the model output by self-feedback correction (Lee et al., 2023; Yin et al., 2023), providing additional knowledge using auxiliary models (Wan et al., 2024; Deng et al., 2024; Zhao et al., 2024; Yang et al., 2024; Kim et al., 2024), and contrastive decoding (Leng et al., 2023; Favero et al., 2024; Zhang et al., 2024; Wang et al., 2024), which refines the model outputs by contrasting the conditional probability of textual responses given the original visual input versus a distorted visual input. Our work adopts a unique approach by applying random image transformations to complement the original image. This provides a wide range of visual contexts, aiming to mitigate hallucinatory visual explanations without the complexities of extra models, additional training, or data requirements.

**Image augmentations for model robustness.** Image augmentations (Shorten & Khoshgoftaar, 2019; Perez, 2017) have long been recognized as a crucial technique for improving model robustness, particularly in computer vision and multimodal tasks. By introducing variations in input data, augmentations help models generalize better to unseen scenarios, reduce overfitting, and improve performance in the presence of noise or ambiguous inputs. In the training phase, data augmentation techniques (Cubuk et al., 2018; Taylor & Nitschke, 2017), such as those used in SimCLR (Chen et al., 2020) and BYOL (Grill et al., 2020), enhance the diversity of training data by applying transformations like rotations, flips, and crops. This encourages the model to learn more generalizable features, improving performance on unseen data. At inference time, test-time augmentation (TTA) (Zhang et al., 2022; Shanmugam et al., 2021; Pérez et al., 2021) further improves model robustness. TTA applies multiple transformations to the input image during testing, generating varied predictions which are then averaged or ensembled to produce a more reliable output. By exposing the model to diverse perspectives of the same input, TTA reduces sensitivity to noise and ambiguity, stabilizes predictions on difficult cases, and serves as a cost-effective ensembling method without requiring additional model training. Our approach builds on these concepts by using random image transformations during inference to provide a broader visual context, reducing hallucinations in vision-language models. By combining predictions from both the original and transformed images, our method enhances robustness without requiring extra training or data.

## 3 APPROACH: 🏯 RITUAL

We present a simple decoding method that can be applied in an online manner during token generation. Our method is training-free, does not require external models or a costly self-feedback mechanism, and remains compatible with existing contrastive decoding techniques (Leng et al., 2023; Favero et al., 2024). An overview of our method is illustrated in Fig. 2.

### 3.1 LVLM FORMULATION

**Vision-Language Alignment.** LVLM takes a visual input and a textual query as inputs, where the visual input provides contextual visual information to assist the model in generating a relevant response to the textual query. Initially, a vision encoder (*e.g.*, ViT (Dosovitskiy et al., 2020), CLIP (Radford et al., 2021), *etc.*) processes raw images to extract visual features. These features are then projected into the language model's input space using a vision-language alignment module (*e.g.*, Q-Former (Li et al., 2023a), linear projection (Liu et al., 2023c), *etc.*), resulting in a set of visual tokens, $\mathcal{V} = \{\nu_0, \nu_1, \ldots, \nu_{N-1}\}$. Concurrently, the textual inputs are tokenized into $\mathcal{T} = \{\tau_N, \tau_{N+1}, \ldots, \tau_{N+M-1}\}$. The visual and textual tokens are concatenated to form an input sequence of length $N + M$.

**Model Forwarding.** The LVLM, parametrized by $\theta$, processes the concatenated sequence of visual and textual tokens. This process is formalized as:

$$\mathcal{H} = \text{LVLM}_\theta([\mathcal{V}, \mathcal{T}]), \tag{1}$$

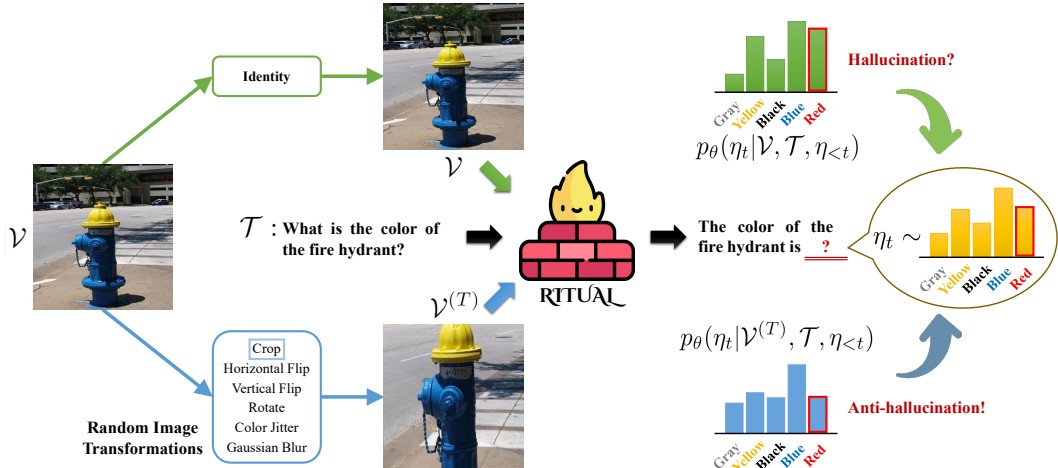

Figure 2: **Overview of 🏯 RITUAL.** At each timestep $t$, LVLM auto-regressively samples a response $\eta_t$ given a visual input, a textual query $\mathcal{T}$, and previously generated tokens $\eta_{<t}$. When conditioned on the original image $\mathcal{V}$, the probabilities for Blue (*correct*) and Red (*hallucinated*) responses are similar, which can lead to the hallucinated response being easily sampled. RITUAL leverages an additional probability distribution conditioned on the transformed image $\mathcal{V}^{(\mathcal{T})}$, where the likelihood of hallucination is significantly reduced. Consequently, the response is sampled from a linear combination of the two probability distributions, ensuring more accurate and reliable outputs.

where $\mathcal{H}$ denotes the sequence of output hidden states from the final layer of LVLM. These hidden states $\mathcal{H}$ are used to compute the logits (or probabilities) for predicting the next tokens.

**Response Generation.** The LVLM generates responses auto-regressively, employing a causal attention mask to ensure each subsequent token is predicted based solely on the preceding tokens. Each response token is generated by sampling from the following probability distribution:

$$\eta_t \sim p_\theta(\eta_t | \mathcal{V}, \mathcal{T}, \eta_{<t}). \tag{2}$$

where $\eta_t$ denotes the response token being generated at timestep $t$, and $\eta_{<t}$ indicates the sequence of tokens generated up to timestep $(t-1)$. This generative process is iteratively continued, appending each newly predicted token to the sequence, until the termination of the sequence. By default, Greedy Decoding is used. Alternatively, decoding strategies such as Beam Search (Wiseman & Rush, 2016), Nucleus Sampling (Holtzman et al., 2019), or DoLa (Chuang et al., 2023) can be employed.

### 3.2 MITIGATING HALLUCINATIONS IN LVLMs WITH RANDOM IMAGE TRANSFORMATIONS

Visual hallucinations in LVLMs can occur during the decoding phase when tokens are selected based on erroneous probability distributions that do not align with the visual inputs. Our approach aims to mitigate these visual hallucinations with a simple yet effective modification to the input handling.

We first randomly apply common image transformations (*e.g.*, Crop, Flip, Color jitter, *etc.*) to the original visual input $\mathcal{V}$, This results in a transformed version of the visual input, $\mathcal{V}^{(\mathcal{T})}$.

$$\mathcal{V}^{(\mathcal{T})} = T(\mathcal{V}; \omega), \text{ where } \omega \in \Omega. \tag{3}$$

Here, $T$ represents a specific transformation function selected randomly from a set of image transformations. The parameter $\omega$ represents the specific parameters of the transformation, drawn from a distribution $\Omega$ that governs the selection and nature of the transformation applied.

During the decoding phase, rather than using $\mathcal{V}^{(\mathcal{T})}$ alone — which we found to impair performance — we utilize both the original and transformed images. This dual-input approach significantly reduces the likelihood of hallucinatory outputs, as illustrated in Fig. 1, and improves the accuracy of the model's predictions. The sampling equation in Eq. (2) is updated as follows:

$$\eta_t \sim p_\theta(\eta_t | \mathcal{V}, \mathcal{T}, \eta_{<t}) + \alpha p_\theta(\eta_t | \mathcal{V}^{(\mathcal{T})}, \mathcal{T}, \eta_{<t}). \tag{4}$$

Here, $\alpha$ is a balancing hyperparameter, adjusting the contribution of the transformed input relative to the original. To promote output diversity and avoid deterministic behavior, we choose to sample from a multinomial distribution rather than merely selecting the most probable output via argmax.

In practice, we employ a predefined set of image transformations to enhance model robustness, divided into geometric and appearance transformations. Geometric transformations, such as flipping, small random rotations, and cropping, simulate different viewing angles, orientations, and focus areas, enhancing the model's ability to generalize across varied perspectives and object positioning. Appearance transformations, including color jitter and Gaussian blur, adjust brightness, contrast, and saturation to account for lighting variations and sensor noise, increasing resilience to image imperfections. Together, these transformations introduce meaningful variations that better prepare the model for real-world image scenarios, improving its flexibility and performance.

## 4 EXPERIMENTS

### 4.1 EVALUATION SETUP

Throughout our experiments, we set hyperparameter configuration at $\alpha = 3$. For random image transformation, we use flip (horizontal & Vertical), rotate, color jitter, Gaussian blur, and crop. In all experimental tables, the *base* refers to the standard decoding, which directly samples the response token from the softmax distribution.[1]

**LVLMs.** We integrate RITUAL with two state-of-the-art LVLMs: **LLaVA-1.5 (** (Liu et al., 2023c) and **InstructBLIP** (Dai et al., 2024). Both models incorporate Vicuna 7B (Chiang et al., 2023) as their language decoding mechanism. LLaVA-1.5 utilizes two-layer MLP to align image and text modalities and InstructBLIP employs the Q-Former (Li et al., 2023a) to efficiently bridge visual and textual features using a fixed number of tokens (*e.g.*, 32). Note that the adaptability of RITUAL extends beyond these two models and is model-agnostic. It can be compatible with a wide range of off-the-shelf LVLMs.

**Baselines.** Our method aims to reduce hallucinations in LVLMs by modifying model's decoding process without relying on external models, costly self-feedback mechanisms, or additional training. To align with these criteria, we select baseline methods that meet these requirements. Recent contrastive decoding methods fit well within this scope, and we establish two primary baselines: **VCD** (Leng et al., 2023) and **M3ID** (Favero et al., 2024). Both VCD and M3ID aim to mitigate object hallucinations by increasing the influence of the reference image over the language prior. This is achieved by contrasting output distributions derived from both original and distorted visual inputs. We also include **DoLa** (Chuang et al., 2023) as a baseline, which employs a novel decoding strategy that contrasts logits from earlier and later layers of the transformer architecture. This amplifies factual knowledge stored in the upper layers while suppressing linguistic patterns from the lower layers that may lead to hallucinations. Additionally, we report results from **OPERA** (Huang et al., 2023), which mitigates hallucinations in LVLMs via an over-trust penalty and retrospection allocation. In contrast to all other methods, OPERA uses beam search during response generation, contributing to its higher performance. We include it for comparison purposes due to its demonstrated effectiveness in reducing hallucinations. All baselines were reproduced within our evaluation setting for consistency.

**Benchmarks.** **(1) POPE** (Li et al., 2023c) frames hallucination assessment as a binary classification task using yes/no questions about object presence (*e.g.*, "Is there a dog in the image?"). It evaluates 500 MS-COCO images with questions based on actual objects or nonexistent objects. The benchmark contains three subsets (random, popular, and adversarial), addressing object prevalence and co-occurrences. **(2) MME** (Fu et al., 2024) is a comprehensive LVLM benchmark assessing 14 subtasks, including object hallucination through tasks like object existence, count, position, and color. These tasks are framed as binary yes/no questions. **(3) CHAIR** (Rohrbach et al., 2018) evaluates the proportion of words in captions that correspond to actual objects in an image, using ground-truth captions and object annotations. It has two variants: (i) per-sentence ($\text{CHAIR}_S$) is defined as $|\{\text{sentences with hallucinated objects}\}|/|\{\text{all sentences}\}|$. (ii) per-instance ($\text{CHAIR}_I$) is defined as $|\{\text{hallucinated objects}\}|/|\{\text{all objects mentioned}\}|$. We randomly select 500 images from the COCO (Lin et al., 2014) validation set and conduct image captioning with the prompt "Please describe this image in detail".

---

[1]We refer readers to Appendix E for further implementation & experimental details and additional results.

Table 1: **Results on POPE (Li et al., 2023c) benchmark.** RITUAL consistently outperforms the contrastive decoding baselines: VCD, M3ID, and DoLa. Moreover, RITUAL is shown to be compatible with both VCD and M3ID, leading to further performance improvements in most configurations. Results are reproduced within our evaluation setting.

| | Setup | Method | LLaVA 1.5 (Liu et al., 2023c) | | | | InstructBLIP (Dai et al., 2024) | | | |
|---|---|---|---|---|---|---|---|---|---|---|
| | | | Acc. ↑ | Prec. ↑ | Rec. ↑ | F1 ↑ | Acc. ↑ | Prec. ↑ | Rec. ↑ | F1 ↑ |
| **MS-COCO (Lin et al., 2014)** | Random | *base* | 84.13 | 82.86 | 86.07 | 84.43 | 82.80 | 82.24 | 83.67 | 82.95 |
| | | VCD | 85.37 | 83.14 | 88.73 | 85.84 | 83.93 | 84.42 | 82.67 | 83.73 |
| | | M3ID | 86.00 | 85.11 | 87.27 | 86.18 | 84.37 | 84.62 | 84.00 | 84.31 |
| | | DoLa | 85.97 | 85.10 | 87.20 | 86.14 | 84.00 | 82.86 | 85.73 | 84.27 |
| | | RITUAL | 88.87 | 89.23 | 88.40 | 88.81 | 88.83 | 90.48 | 86.80 | 88.60 |
| | | RITUAL+VCD | 89.07 | 89.49 | 88.53 | 89.01 | 89.30 | 90.85 | 87.40 | 89.09 |
| | | RITUAL+M3ID | 89.00 | 89.85 | 87.93 | 88.88 | 88.93 | 91.13 | 86.27 | 88.63 |
| | | OPERA (Beam) | 89.37 | 92.03 | 86.20 | 89.02 | 89.17 | 95.51 | 82.20 | 88.36 |
| | Popular | *base* | 80.87 | 78.23 | 85.53 | 81.72 | 75.80 | 72.74 | 82.53 | 77.33 |
| | | VCD | 81.10 | 77.78 | 87.07 | 82.16 | 77.73 | 75.43 | 82.27 | 78.70 |
| | | M3ID | 82.83 | 79.62 | 88.27 | 83.72 | 77.30 | 74.10 | 83.93 | 78.71 |
| | | DoLa | 82.93 | 79.76 | 88.27 | 83.80 | 77.37 | 73.50 | 85.60 | 79.09 |
| | | RITUAL | 85.83 | 84.17 | 88.27 | 86.17 | 81.97 | 78.90 | 87.27 | 82.87 |
| | | RITUAL+VCD | 85.77 | 83.89 | 88.53 | 86.15 | 82.83 | 80.16 | 87.27 | 83.56 |
| | | RITUAL+M3ID | 85.37 | 83.60 | 88.00 | 85.74 | 81.90 | 78.98 | 86.93 | 82.77 |
| | | OPERA (Beam) | 86.20 | 85.17 | 87.67 | 86.40 | 84.07 | 85.39 | 82.20 | 83.76 |
| | Adversarial | *base* | 76.23 | 71.75 | 86.53 | 78.45 | 75.40 | 71.60 | 84.20 | 77.39 |
| | | VCD | 75.60 | 70.78 | 87.20 | 78.14 | 76.80 | 73.62 | 83.53 | 78.26 |
| | | M3ID | 77.70 | 73.23 | 87.33 | 79.66 | 76.03 | 72.48 | 83.93 | 77.79 |
| | | DoLa | 77.17 | 72.30 | 88.07 | 79.41 | 74.30 | 69.95 | 85.20 | 76.83 |
| | | RITUAL | 78.80 | 74.43 | 87.73 | 80.54 | 78.73 | 74.57 | 87.20 | 80.39 |
| | | RITUAL+VCD | 79.60 | 75.26 | 88.20 | 81.22 | 79.07 | 74.89 | 87.47 | 80.69 |
| | | RITUAL+M3ID | 79.20 | 74.83 | 88.00 | 80.88 | 78.93 | 75.06 | 86.67 | 80.45 |
| | | OPERA (Beam) | 81.07 | 77.44 | 87.67 | 82.24 | 81.83 | 81.60 | 82.20 | 81.90 |
| **A-OKVQA (Schwenk et al., 2022)** | Random | *base* | 81.73 | 76.53 | 91.53 | 83.36 | 81.13 | 78.03 | 86.67 | 82.12 |
| | | VCD | 81.83 | 75.74 | 93.67 | 83.76 | 82.00 | 79.38 | 86.47 | 82.77 |
| | | M3ID | 83.57 | 77.86 | 93.80 | 85.09 | 82.33 | 77.81 | 90.47 | 83.66 |
| | | DoLa | 83.23 | 77.47 | 93.73 | 84.83 | 82.17 | 78.17 | 89.27 | 83.35 |
| | | RITUAL | 85.17 | 79.79 | 94.20 | 86.40 | 87.13 | 83.92 | 91.87 | 87.71 |
| | | RITUAL+VCD | 85.10 | 79.93 | 93.73 | 86.28 | 86.77 | 83.57 | 91.53 | 87.37 |
| | | RITUAL+M3ID | 85.93 | 80.62 | 94.60 | 87.06 | 87.17 | 84.35 | 91.27 | 87.67 |
| | | OPERA (Beam) | 86.80 | 82.90 | 92.73 | 87.54 | 89.97 | 90.75 | 89.00 | 89.87 |
| | Popular | *base* | 76.67 | 70.51 | 91.67 | 79.71 | 75.67 | 70.97 | 86.87 | 78.12 |
| | | VCD | 74.70 | 68.12 | 92.87 | 78.59 | 76.50 | 71.69 | 87.60 | 78.85 |
| | | M3ID | 76.80 | 70.20 | 93.13 | 80.06 | 75.60 | 70.40 | 88.33 | 78.36 |
| | | DoLa | 76.47 | 69.79 | 93.33 | 79.86 | 76.93 | 71.15 | 90.60 | 79.71 |
| | | RITUAL | 78.83 | 71.99 | 94.40 | 81.68 | 78.73 | 72.83 | 91.67 | 81.17 |
| | | RITUAL+VCD | 79.17 | 72.40 | 94.27 | 81.90 | 78.83 | 72.75 | 92.20 | 81.33 |
| | | RITUAL+M3ID | 79.63 | 72.83 | 94.27 | 82.27 | 79.20 | 73.42 | 91.53 | 81.48 |
| | | OPERA (Beam) | 79.60 | 73.44 | 92.73 | 81.97 | 82.60 | 78.90 | 89.00 | 83.65 |
| | Adversarial | *base* | 67.40 | 61.78 | 91.27 | 73.68 | 68.00 | 63.08 | 86.80 | 73.06 |
| | | VCD | 67.43 | 61.48 | 93.33 | 74.13 | 70.67 | 65.24 | 88.47 | 75.10 |
| | | M3ID | 68.10 | 61.99 | 93.60 | 74.58 | 69.57 | 64.21 | 88.40 | 74.39 |
| | | DoLa | 68.03 | 62.02 | 93.07 | 74.43 | 68.50 | 62.94 | 90.00 | 74.07 |
| | | RITUAL | 68.57 | 62.26 | 94.27 | 74.99 | 70.27 | 64.15 | 91.87 | 75.55 |
| | | RITUAL+VCD | 68.80 | 62.48 | 94.13 | 75.11 | 71.00 | 64.72 | 92.33 | 76.10 |
| | | RITUAL+M3ID | 68.77 | 62.42 | 94.33 | 75.13 | 69.30 | 63.43 | 91.13 | 74.80 |
| | | OPERA (Beam) | 70.00 | 63.75 | 92.73 | 75.56 | 74.53 | 69.03 | 89.00 | 77.75 |
| **GQA (Hudson & Manning, 2019)** | Random | *base* | 81.23 | 75.42 | 92.67 | 83.16 | 79.93 | 76.73 | 85.93 | 81.07 |
| | | VCD | 81.50 | 74.78 | 95.07 | 83.71 | 81.83 | 79.03 | 86.67 | 82.67 |
| | | M3ID | 82.83 | 76.64 | 94.47 | 84.62 | 80.57 | 76.77 | 87.67 | 81.85 |
| | | DoLa | 83.70 | 77.70 | 94.53 | 85.29 | 81.57 | 77.90 | 88.13 | 82.70 |
| | | RITUAL | 86.10 | 80.30 | 95.67 | 87.31 | 84.87 | 82.52 | 88.47 | 85.39 |
| | | RITUAL+VCD | 86.03 | 80.21 | 95.67 | 87.26 | 84.97 | 82.40 | 88.93 | 85.54 |
| | | RITUAL+M3ID | 86.30 | 80.64 | 95.53 | 87.46 | 85.00 | 82.94 | 88.13 | 85.46 |
| | | OPERA (Beam) | 87.07 | 82.25 | 94.53 | 87.97 | 87.70 | 90.02 | 84.80 | 87.33 |
| | Popular | *base* | 72.50 | 65.85 | 93.47 | 77.27 | 72.73 | 68.14 | 85.40 | 75.80 |
| | | VCD | 71.57 | 64.72 | 94.80 | 76.93 | 73.67 | 68.82 | 86.53 | 76.67 |
| | | M3ID | 72.83 | 66.04 | 94.00 | 77.58 | 74.57 | 69.45 | 87.73 | 77.53 |
| | | DoLa | 74.03 | 66.85 | 95.33 | 78.59 | 73.70 | 68.58 | 87.47 | 76.88 |
| | | RITUAL | 74.80 | 67.50 | 95.67 | 79.15 | 74.50 | 69.17 | 88.40 | 77.61 |
| | | RITUAL+VCD | 75.07 | 67.82 | 95.40 | 79.28 | 75.33 | 69.98 | 88.73 | 78.25 |
| | | RITUAL+M3ID | 74.40 | 67.15 | 95.53 | 78.87 | 75.57 | 70.24 | 88.73 | 78.41 |
| | | OPERA (Beam) | 75.50 | 68.47 | 94.53 | 79.42 | 78.77 | 75.67 | 84.80 | 79.97 |
| | Adversarial | *base* | 67.63 | 61.68 | 93.13 | 74.21 | 69.57 | 64.80 | 85.67 | 73.79 |
| | | VCD | 67.47 | 61.38 | 94.20 | 74.33 | 69.43 | 64.76 | 85.27 | 73.61 |
| | | M3ID | 68.13 | 61.88 | 94.47 | 74.78 | 68.90 | 64.06 | 86.13 | 73.47 |
| | | DoLa | 68.73 | 62.34 | 94.67 | 75.17 | 69.70 | 64.28 | 88.67 | 74.53 |
| | | RITUAL | 68.23 | 61.75 | 95.80 | 75.10 | 70.17 | 64.76 | 88.47 | 74.78 |
| | | RITUAL+VCD | 69.00 | 62.39 | 95.67 | 75.53 | 70.23 | 64.81 | 88.53 | 74.84 |
| | | RITUAL+M3ID | 68.80 | 62.29 | 95.27 | 75.33 | 71.00 | 65.32 | 89.53 | 75.53 |
| | | OPERA (Beam) | 70.00 | 63.42 | 94.53 | 75.91 | 74.40 | 70.20 | 84.80 | 76.81 |

Table 2: **Results on MME-Hallucination (Fu et al., 2024).** RITUAL effectively mitigates hallucinations at both the object and attribute levels, outperforming contrastive decoding methods in Total Score.

| Model | Method | Object-level | | Attribute-level | | Total Score |
|---|---|---|---|---|---|---|
| | | Existence ↑ | Count ↑ | Position ↑ | Color ↑ | |
| LLaVA 1.5 | *base* | $173.75_{(\pm 4.79)}$ | $121.67_{(\pm 12.47)}$ | $117.92_{(\pm 3.69)}$ | $149.17_{(\pm 7.51)}$ | $562.50_{(\pm 3.96)}$ |
| | VCD | $178.75_{(\pm 2.50)}$ | $126.25_{(\pm 10.40)}$ | $120.00_{(\pm 4.08)}$ | $150.83_{(\pm 11.01)}$ | $575.84_{(\pm 9.67)}$ |
| | M3ID | $177.50_{(\pm 6.45)}$ | $124.17_{(\pm 10.93)}$ | $120.00_{(\pm 7.07)}$ | $152.92_{(\pm 5.67)}$ | $574.59_{(\pm 9.75)}$ |
| | DoLa | $174.58_{(\pm 5.34)}$ | $122.09_{(\pm 11.73)}$ | $122.09_{(\pm 2.10)}$ | $149.17_{(\pm 4.19)}$ | $567.92_{(\pm 13.63)}$ |
| | RITUAL | $187.50_{(\pm 2.89)}$ | $139.58_{(\pm 7.62)}$ | $125.00_{(\pm 10.27)}$ | $164.17_{(\pm 6.87)}$ | $616.25_{(\pm 20.38)}$ |
| | RITUAL+VCD | $185.00_{(\pm 4.08)}$ | $140.84_{(\pm 4.41)}$ | $125.00_{(\pm 7.07)}$ | $165.83_{(\pm 6.46)}$ | $616.67_{(\pm 11.14)}$ |
| | RITUAL+M3ID | $187.50_{(\pm 2.89)}$ | $141.25_{(\pm 9.85)}$ | $125.00_{(\pm 10.27)}$ | $164.17_{(\pm 6.87)}$ | $617.92_{(\pm 22.12)}$ |
| InstructBLIP | *base* | $160.42_{(\pm 5.16)}$ | $79.17_{(\pm 8.22)}$ | $79.58_{(\pm 8.54)}$ | $130.42_{(\pm 17.34)}$ | $449.58_{(\pm 24.09)}$ |
| | VCD | $158.75_{(\pm 7.25)}$ | $90.75_{(\pm 3.11)}$ | $70.00_{(\pm 15.81)}$ | $132.50_{(\pm 18.78)}$ | $452.00_{(\pm 31.57)}$ |
| | M3ID | $158.33_{(\pm 5.44)}$ | $94.58_{(\pm 9.85)}$ | $72.50_{(\pm 17.03)}$ | $128.33_{(\pm 14.72)}$ | $453.75_{(\pm 26.82)}$ |
| | DoLa | $162.08_{(\pm 5.34)}$ | $82.50_{(\pm 6.16)}$ | $78.75_{(\pm 8.96)}$ | $135.42_{(\pm 10.49)}$ | $458.75_{(\pm 11.25)}$ |
| | RITUAL | $182.50_{(\pm 6.45)}$ | $74.58_{(\pm 5.99)}$ | $67.08_{(\pm 10.31)}$ | $139.17_{(\pm 0.96)}$ | $463.33_{(\pm 12.40)}$ |
| | RITUAL+VCD | $185.00_{(\pm 4.08)}$ | $75.00_{(\pm 7.07)}$ | $62.50_{(\pm 6.46)}$ | $141.67_{(\pm 6.53)}$ | $464.17_{(\pm 9.07)}$ |
| | RITUAL+M3ID | $182.50_{(\pm 6.45)}$ | $74.58_{(\pm 2.84)}$ | $63.33_{(\pm 11.55)}$ | $140.42_{(\pm 2.10)}$ | $460.83_{(\pm 11.1)}$ |

Figure 3: **Comparison on MME-Fullset (Fu et al., 2024).** When equipped with RITUAL, LLaVA-1.5 (Liu et al., 2023c) performs best in 12 out of 14 categories, while InstructBLIP (Dai et al., 2024) excels in 8 categories. RITUAL not only reduces hallucinations but also enhances the general capabilities of LVLMs. Detailed results are in Appendix F.4.

## 4.2 RESULTS

**Results on POPE.** Table 1 compares various decoding-based hallucination mitigation methods on the POPE benchmark (Li et al., 2023c), evaluated with two representative LVLMs: LLaVA 1.5 (Liu et al., 2023c) and InstructBLIP (Dai et al., 2024). The results demonstrate that RITUAL consistently outperforms baseline, VCD (Leng et al., 2023), M3ID (Favero et al., 2024), and DoLa (Chuang et al., 2023) across all datasets (MS-COCO (Lin et al., 2014), A-OKVQA (Schwenk et al., 2022), and GQA (Hudson & Manning, 2019)) and setups (random, popular, and adversarial), and all metrics, demonstrating its robustness in mitigating hallucinations. This underscores the importance of considering visual context from multiple perspectives. Furthermore, RITUAL yields further performance improvement when incorporated with contrastive decoding methods (VCD and M3ID), indicating compatibility. This synergy between contrastive decoding, which aims to reduce language biases, and our approach, which captures a broader range of visual contexts through varying fields of view, effectively mitigates object hallucinations.

**Results on MME-Hallucination.** In Table 2, we compare the results on the MME-hallucination subset (Fu et al., 2024) to verify the model's effectiveness in reducing various types of hallucinations beyond object existence. When combined with LLaVA-1.5 (Liu et al., 2023c), RITUAL outperforms all counterparts across both object-level (Existence and Count) and attribute-level (Position and Color) evaluations. With InstructBLIP (Dai et al., 2024), while the other methods show a slight advantage

Table 3: **Results on CHAIR (Rohrbach et al., 2018) benchmark.** RITUAL significantly reduces object hallucinations in caption generation compared to VCD, M3ID, and DoLa. It can also boost performance when combined with VCD and M3ID. The number of *max new tokens* is set to 64.

| | Method | $CHAIR_S\downarrow$ | $CHAIR_I\downarrow$ |
|---|---|---|---|
| LLaVA 1.5 | *base* | 26.2 | 9.3 |
| | VCD | 22.4 | 7.6 |
| | M3ID | 23.0 | 6.8 |
| | DoLa | 23.2 | 7.8 |
| | RITUAL | 20.6 | 6.9 |
| | RITUAL+VCD | 20.0 | 6.8 |
| | RITUAL+M3ID | 18.0 | 5.7 |
| | OPERA (beam) | 23.0 | 7.5 |
| InstructBLIP | *base* | 28.6 | 10.3 |
| | VCD | 27.2 | 9.1 |
| | M3ID | 31.8 | 10.4 |
| | DoLa | 36.6 | 12.5 |
| | RITUAL | 26.0 | 8.8 |
| | RITUAL+VCD | 25.0 | 8.6 |
| | RITUAL+M3ID | 23.4 | 7.9 |
| | OPERA (beam) | 25.6 | 8.3 |

Table 4: **Generated Text Quality.** RITUAL demonstrates a competitive level of text quality compared to other decoding methods.

| Method | Grammar $\uparrow$ | Fluency $\uparrow$ |
|---|---|---|
| *base* | 9.804 | 9.432 |
| VCD | 9.802 | 9.352 |
| M3ID | 9.832 | 9.344 |
| DoLa | 9.814 | 9.320 |
| RITUAL | 9.844 | 9.398 |
| OPERA | 9.828 | 9.308 |

Table 5: **Comparison of performance and latency on COCO random setup.**

| Method | LLaVA 1.5 | | | | |
|---|---|---|---|---|---|
| | Acc. $\uparrow$ | Prec. $\uparrow$ | Rec. $\uparrow$ | F1 $\uparrow$ | Latency (ms/token) |
| *base* | 84.13 | 82.86 | 86.07 | 84.43 | 21.96 |
| VCD | 85.37 | 83.14 | 88.73 | 85.84 | 43.33 |
| M3ID | 86.00 | 85.11 | 87.27 | 86.18 | 40.07 |
| DoLa | 85.97 | 85.10 | 87.20 | 86.14 | 28.70 |
| RITUAL | 88.87 | 89.23 | 88.40 | 88.81 | 43.37 |
| OPERA (beam) | 89.37 | 92.03 | 86.20 | 89.02 | 308.48 |

in Count and Position, RITUAL surpasses the baseline and other contrastive decoding methods in the total score. Moreover, when combined with existing methods like VCD (Leng et al., 2023) and M3ID, RITUAL exhibits further performance enhancement. RITUAL exhibits lower performance in Count and Position tasks due to the inherent challenges associated with specific transformations. For instance, tasks like Count may be impacted by cropping transformations that alter the visible quantity of objects, while Position accuracy may be affected by flipping transformations that change the spatial arrangement of objects.

**Results on MME-Fullset.** As depicted in Fig. 3, we evaluate the MME-Fullset (Fu et al., 2024) to assess the impact of decoding methods on the general ability of LVLMs. Across 14 categories, both LLaVA-1.5 and InstructBLIP adopting RITUAL consistently achieve the highest scores across most tasks, demonstrating its effectiveness of RITUAL in improving visual and textual understanding. By enriching the model's visual capacity from diverse visual contexts, RITUAL provides a balanced enhancement across a wide range of tasks, making it a versatile and robust method for improving LVLM performance. Despite these advancements, some tasks may still exhibit lower performance due to the inherent challenges of statistical bias and language priors affecting LVLMs.

**Results on CHAIR.** To assess the reduction of object existence hallucination, we use the CHAIR metrics, where the presence of objects in the description serves as the measurement criterion. Given the generative nature of the task, we limit the maximum number of new tokens to 64. As shown in Table 3, our RITUAL outperforms both the baseline and previous contrastive decoding approaches. For LLaVA 1.5, RITUAL achieves $CHAIR_S$ and $CHAIR_I$ scores of 20.6 and 6.9, respectively, significantly surpassing both baseline and VCD. While M3ID shows slightly better performance in $CHAIR_I$, RITUAL attains comparable scores and markedly excels in $CHAIR_I$. Similarly, for InstructBLIP, RITUAL achieves the best results with $CHAIR_S$ and $CHAIR_I$ scores of 26.0 and 8.8, respectively. Additionally, when combined with VCD and M3ID, RITUAL further reduces the CHAIR score.

### 4.3 ANALYSIS

**Generation Quality.** Since previous methods and RITUAL modify the logits from the standard decoding strategy, there may be concerns about potentially compromising the quality of the generated text. Therefore, we employed GPT-4-Turbo to evaluate the grammar and fluency of generated text from 500 samples of the CHAIR benchmark using the InstructBLIP. As shown in the Tab. 4, our

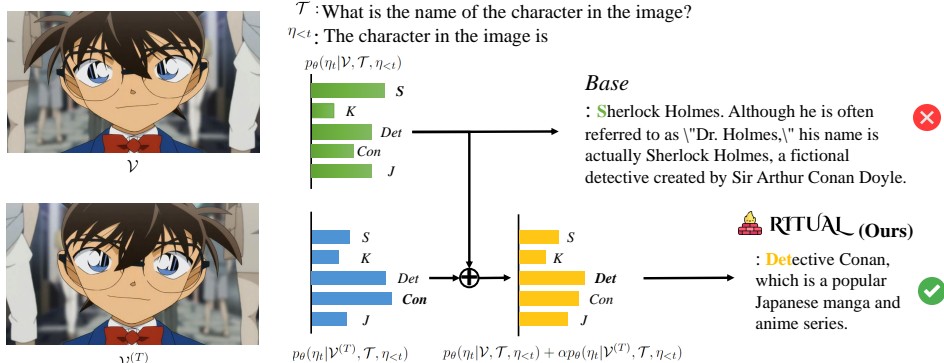

Figure 4: RITUAL **in descriptive task.** RITUAL refines the probability distribution to generate the correct token during decoding, thereby mitigating hallucinations in the following generated text.

decoding method demonstrates text generation quality that is comparable to or exceeds that of the previous work in terms of grammar and fluency. The results highlight the robustness and effectiveness of our method in generating grammatically correct and fluent text while also improving hallucination mitigation without compromising overall text generation quality.

**Latency.** Contrastive decoding methods like VCD and M3ID, as well as RITUAL, require performing the forward process twice to compare two probability distributions, doubling resource consumption. Tab. 5 details the performance and speed comparison. In our experiments, DoLa has minimal overhead compared to normal decoding, with only a 1.3x increase in latency. DoLa is faster than RITUAL, but RITUAL shows better performance. Despite implementation differences such as beam search, OPERA achieves slightly higher accuracy than RITUAL, but our method is significantly faster than OPERA. There are trade-offs among the methods, but RITUAL offers clear advantages. It is conceptually and implementation-wise simple, applicable to various methods, and delivers a favorable speed and performance trade-off. Also, it can be complementarily used with other contrastive decoding methods.

RITUAL **in descriptive task.** We demonstrate how RITUAL is effective in descriptive tasks such as CHAIR in Figure 4. In the case of standard decoding, the model assigns the highest probability to the token 'S' at the current timestep $t$, leading to the incorrect prediction of "Sherlock Holmes." In contrast, RITUAL, which utilizes both the original and augmented images, effectively adjusts the probability distribution and selects the token '$Det$' rather than '$S$', resulting in the correct prediction of "Detective Conan." This highlights the advantage of leveraging augmented images for probability correction, thereby improving accuracy in visually ambiguous contexts.

**Ablation of the number of augmented images.** To investigate whether increased exposure to diverse visual scenarios allows the model to better understand images and produce more robust responses, we conducted an ablation study by varying the number of augmented images in RITUAL. As shown in Tab. 6, the performance slightly improves as more augmented images are used. This improvement can be attributed to the richer visual context provided by the additional augmentations. However, using multiple augmented images also introduces a trade-off, as it increases latency due to the additional computational load. Detailed results are in Appendix F.6.

Table 6: **Ablation of the number of augmented images in** RITUAL **on COCO random.**

| # of Aug. Images | LLaVA-1.5 | |
|---|---|---|
| | Acc. ↑ | F1 ↑ |
| 1 | 88.87 | 88.81 |
| 2 | 89.07 | 89.02 |
| 3 | 89.17 | 89.16 |

**Qualitative results on LLaVA-Bench.** Fig. 5 presents two samples from the LLaVA-Bench (Liu et al., 2023c) with LLaVa-1.5 (Liu et al., 2023c), highlighting the differences between sentences generated by standard decoding (Base) and those produced by RITUAL. The results demonstrate that standard decoding often results in hallucinations, which can be effectively rectified by implementing RITUAL. For instance, in the left-hand image, the baseline model incorrectly identifies a 'street vendor' and 'initiative signs', neither of which are present in the image. Additionally, it misinterprets 'ironing' as 'doing laundry'. In the right-hand image, the baseline model hallucinates objects not present in the image, such as a 'hat', 'paint mustache', and 'two more dogs'. In contrast, our approach helps counteract these hallucinations, generating sentences that reflect a more accurate comprehension of the image.

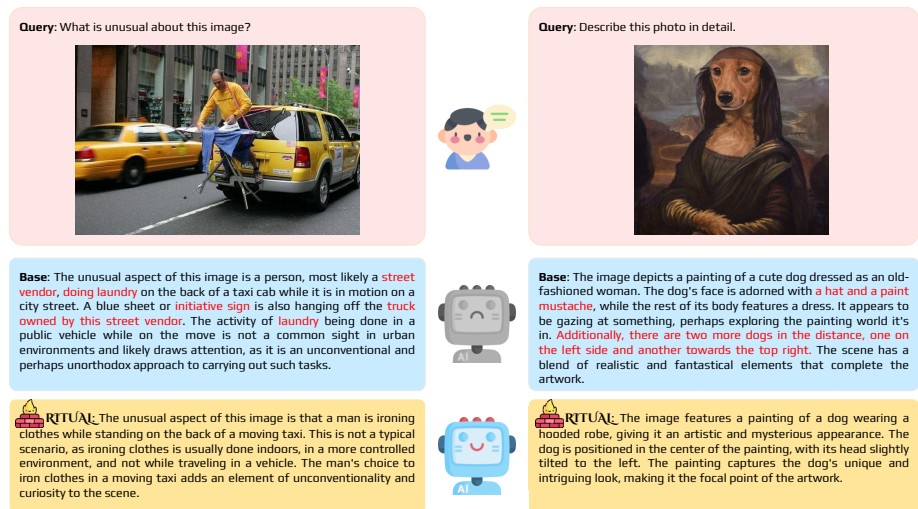

Figure 5: **Results on LLaVA-Bench (Liu et al., 2023c).** Hallucinations are highlighted in red. RITUAL well understands ambiguous images and effectively mitigates hallucinations in outputs.

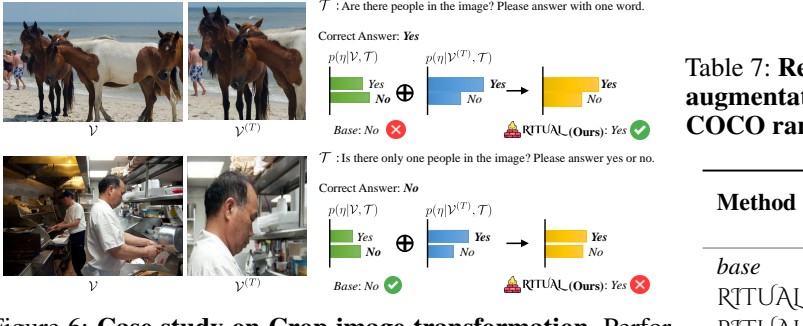

Figure 6: **Case study on Crop image transformation.** Performance can be affected by the cropping area. The randomness of the selected region may sometimes lead to poor outcomes.

Table 7: **Results of self-feedback augmentation selection on COCO random setup.**

| Method | LLaVA 1.5 | |
|---|---|---|
| | Acc. ↑ | F1 ↑ |
| *base* | 84.13 | 84.43 |
| RITUAL | 88.87 | 88.81 |
| RITUAL+ | 89.17 | 89.21 |

## 5 DISCUSSION

In this study, we have introduces RITUAL, a simple approach aimed at enhancing the reliability of LVLMs. We found that while relying solely on random image transformations can degrade performance, they contribute to mitigating hallucination when used in combination with the original image. Inspired by these findings, RITUAL employs random image transformations to provide LVLMs with a broader visual context, thereby improving the model's robustness against hallucinatory outputs. RITUAL significantly outperforms existing approaches on multiple hallucination benchmarks without requiring additional model training or complex external mechanisms. Moreover, RITUAL is also compatible with existing contrastive decoding techniques, further enhancing performance.

**Case Study & Limitations.** As shown in Fig. 6, the effectiveness of specific transformations, such as cropping, can depend heavily on the nature of the query. Cropping might adjust the position of critical spatial regions, enhancing relevance in the above case while detracting from it in the below case. To illustrate this point, while certain transformations might excel under particular conditions, their efficacy can diminish in others. To mitigate this variability, we opt for a randomized selection from a pool of transformations, allowing for a broader range of adaptability across different image and query contexts. Recognizing the need for a more tailored approach, we introduce a self-feedback mechanism, referred to as RITUAL+, which dynamically selects image transformations that are aware of the image-query context. As shown in Tab. 7, this method demonstrates a modest improvement in performance by aligning transformations more closely with the specifics of each query. Implementation details and detailed results are in Appendix F.2. In future work, we aim to develop a more sophisticated mechanism that can more effectively determine the most suitable transformations based on the interplay between the image and its associated query. [2]

---

[2]Additional case studies can be found in Appendix F.10.

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

# APPENDIX

## CONTENTS

## A   LARGE VISION LANGUAGE MODELS (LVLMS)

Recent approaches to integrating visual and language modalities in LVLMs commonly leverage pre-trained uni-modal models. They include an adaptive interface to bridge pre-trained visual encoders with Large Language Models (LLMs), facilitating efficient information synthesis across modalities. These interfaces generally fall into two main categories: (1) *Learnable query-based methods*, exemplified by Q-Former (Li et al., 2023a) in InstructBLIP (Dai et al., 2024) and MiniGPT-4 (Zhu et al., 2023), a set of learnable query tokens is employed to capture visual signals through cross-attention. These tokens are optimized to distill the essential visual information and input it into the LLM for further processing. (2) *Projection layer-based methods*, such as LLaVA (Liu et al., 2023c;b) and Shikra (Chen et al., 2023), use projection layers to transform visual features into the input space of LLMs. This mapping ensures seamless integration between pre-trained visual representations and the LLMs, enabling the latter to interpret the visual content effectively. Both strategies translate visual features into formats that the LLMs can understand. Despite their efficacy, LVLMs still encounter challenges with hallucination, which we aim to mitigate in this work. We specifically use two representative models, LLAVA and InstructBLIP, for experiments.

## B   COMPARISON TO CONTRASTIVE DECODING

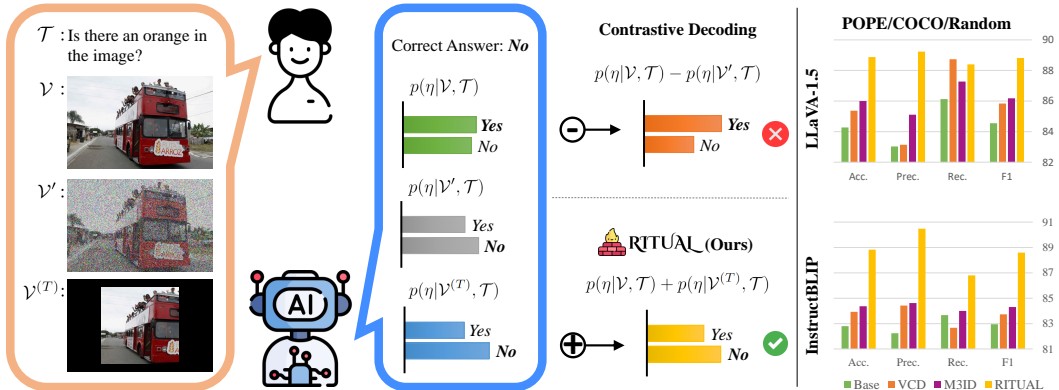

Figure 7: **Comparison of 🏮 RITUAL with contrastive decoding.** Unlike contrastive decoding methods (Leng et al., 2023; Favero et al., 2024), which contrast the conditional probability given the original image ($\mathcal{V}$) to that given a diffused (Leng et al., 2023) (or absent (Favero et al., 2024)) image ($\mathcal{V}'$), we leverage both the original image ($\mathcal{V}$) and a randomly transformed image ($\mathcal{V}^{(T)}$) in a complementary manner. While simple, RITUAL achieves state-of-the-art performance on multiple hallucination benchmarks.

Contrastive decoding (Leng et al., 2023; Favero et al., 2024; Zhang et al., 2024; Wang et al., 2024) refines the model outputs by contrasting the conditional probability of textual responses given the original visual input versus a distorted visual input. This method aims to alleviate language biases or statistical priors, ensuring that responses are more grounded in the actual images, thereby reducing deviations from the visual truth. While beneficial, contrastive decoding does not fully resolve the misalignments between visual data and textual descriptions and can sometimes lead to the reinforcement of incorrect patterns.

Our method is distinct from contrastive decoding (Leng et al., 2023; Favero et al., 2024; Wang et al., 2024), which attributes the causes of hallucinations to language bias or statistical priors. Instead, RITUAL suggests that the source of hallucinatory content might actually reside within the images themselves, advocating for a multifaceted view of visual inputs. The conceptual comparison is shown in Fig. 7.

## C   ADDITIONAL CONTEXT ON TEST-TIME AUGMENTATION

Test-Time Augmentation (TTA) is a technique designed to improve model robustness and generalization during inference by using multiple augmented versions of an input. By applying transformations

such as rotations, flips, or crops, TTA reduces uncertainty and enhances accuracy through prediction averaging or ensembling across these variations. This is particularly useful for tasks with high input variability or noise, as it allows the model to handle perturbations that might otherwise degrade performance.

TTA works by exposing the model to different transformations of the same image, enabling it to make predictions for each variation as well as the original input. These predictions are then aggregated to produce a more stable and reliable final output, effectively mitigating the impact of noisy or ambiguous test data. This process also helps stabilize predictions in cases where the input lies near a decision boundary, offering a more balanced perspective by incorporating diverse views of the image.

An additional advantage of TTA is that it serves as a lightweight ensembling method. While traditional ensembling requires training multiple models, TTA leverages a single model to generate predictions on different augmented versions of the input. This approach achieves the benefits of an ensemble without the computational overhead, making it a cost-effective solution.

Our method builds upon this foundation by applying simple random transformations—such as rotations, flips, or noise—during inference. These augmentations provide the model with a broader visual context, allowing it to capture a wider range of potential interpretations while reducing the risk of hallucinated outputs. By combining the predictions from both the original and transformed images, we enhance the model's robustness without requiring additional training or complex architectures.

# D DETAILED EXPERIMENTAL SETTINGS

**POPE**[3]

We utilize the official benchmark from (Li et al., 2023c), which includes 3,000 question-answer pairs for each of the random, popular, and adversarial settings. We use the query template 'Is there a [object] in the image?'. Here, [object] is selected randomly, from the most frequent objects in the dataset, or from objects that frequently co-occur with [object], corresponding to the random, popular, and adversarial settings respectively. We evaluate the performance based on whether the model-generated output contained the ground truth ('Yes' or 'No') using accuracy, precision, recall, and average F1-score.

**MME**[4]

The MME (Fu et al., 2024) dataset consists of 10 perception categories (existence, count, position, color, posters, celebrity, scene, landmark, artwork, OCR) and 4 recognition ones (commonsense reasoning, numerical calculation, text translation, code reasoning). Each query is used with an image-related question followed by 'Please answer yes or no.'" We report the sum of accuracy at the query level and image level following the official implementation.

**CHAIR**[5]

We select 500 random images from the COCO (Lin et al., 2014) validation set and generate the output using the query "Please Describe this image in detail.". Due to the computational complexity, we restrict the *max new tokens* to 64. Following the M3ID (Favero et al., 2024), we report two assessment metrics, $C_s$ and $C_i$, which calculate the hallucination ratio per sentence and instance as follows:

$$C_s = \frac{|\{\text{sentences with hallucinated objects}\}|}{|\{\text{all sentences}\}|}, C_i = \frac{|\{\text{hallucinated objects}\}|}{|\{\text{all objects mentioned}\}|}. \quad (5)$$

**LLaVA-Bench**[6]

The LLaVA-Bench (Liu et al., 2023c) dataset consists of 24 images along with 60 image-related questions. This dataset is demanding as it has been collected from a variety of domains including diverse scenes, memes, paintings, sketches, and more. We conduct qualitative case studies on this dataset to exhibit the efficacy of RITUAL in challenging tasks and its adaptability to new domains.

---

[3]https://github.com/RUCAIBox/POPE

[4]https://github.com/BradyFU/Awesome-Multimodal-Large-Language-Models/tree/Evaluation

[5]https://github.com/LisaAnne/Hallucination

[6]https://huggingface.co/datasets/liuhaotian/llava-bench-in-the-wild

# E  FURTHER IMPLEMENTATION DETAILS

## E.1  IMAGE TRANSFORMATION

We set predefined six commonly used image transformations and randomly applied one of them for each image. We provide a concise description and implementation details below. We employ the Pytorch/Torchvision (Paszke et al., 2019) implementation for transformation.

**Horizontal flip.** Flip the image in the horizontal direction.

**Vertical flip.** Flip the image in the vertical direction.

**Rotate.** Rotate the image by angle. We set *degrees*=$(-180, +180)$.

**Color jitter.** Change the brightness, contrast, saturation, and hue of an image. We set *brightness*=1, *contrast*=1, *saturation*=1, *hue*=0.5.

**Gaussian blur.** Blurs image with randomly chosen Gaussian blur. We set *kernel_size*=13 and *sigma*=(1.5, 2.0).

**Crop.** Crop a random portion of an image and resize it to a given size. We set *size*=336 as the same as the original data resize scale.

## E.2  DECODING METHODS

For a fair comparison, we adopt adaptive plausible constraint based on the confidence level related to the output distribution from the original visual inputs following (Li et al., 2022; Leng et al., 2023).

$$\mathcal{O}(\eta_{<t}) = \{\eta_t \in \mathcal{O} : p_\theta(\eta_t \mid \mathcal{V}, \mathcal{T}, \eta_{<t}) \geq \beta \max_w p_\theta(w \mid v, x, y_{<t})\}. \tag{6}$$

where $\mathcal{O}$ is the output vocabulary of LVLM, and $\beta$ is a plausible constraint parameter hyperparameter that adjusts the truncation of the next token distribution. The logits of tokens not in $\mathcal{O}$ are set $-\infty$ so that larger $\beta$ results in retaining only tokens with higher probabilities. We set $\beta$= 0.1 for all experiments. We configured the hyperparameter with a value of $\alpha = 3$ in Eq. (4) by default. Note that we reproduced VCD (Leng et al., 2023) and M3ID (Favero et al., 2024) with our settings. We use the contrastive distribution of VCD as shown in Eq. (7) and set the balancing parameter $\gamma$=2 and $\delta$=1, and the total noise step = 500 for generating the corrupted image $\mathcal{V}'$.

$$\eta_t^{VCD} \sim \gamma p_\theta(\eta_t|\mathcal{V}, \mathcal{T}, \eta_{<t}) - \delta p_\theta(\eta_t|\mathcal{V}', \mathcal{T}, \eta_{<t}). \tag{7}$$

Furthermore, we reproduced a key concept of M3ID, preventing conditioning dilution by introducing the unconditioned model as below:

$$\eta_t^{M3ID} \sim p_\theta(\eta_t|\mathcal{V}, \mathcal{T}, \eta_{<t}) + \frac{1 - e^{-\lambda t}}{e^{-\lambda t}}(p_\theta(\eta_t|\mathcal{V}, \mathcal{T}, \eta_{<t}) - p_\theta(\eta_t|\mathcal{T}, \eta_{<t})) \tag{8}$$

We set the $\lambda$, balancing parameter between conditioned model and unconditioned model, to 0.1. Note that $\eta_t^{Transformed} = p_\theta(\eta_t|\mathcal{V}^{(T)}, \mathcal{T}, \eta_{<t})$. When we use RITUAL and contrastive decoding, we used combined distribution as $\zeta \eta_t^{Transformed} + \eta_t^D$ where $\{VCD, M3ID\} \in D$. In this case, we set $\gamma$=1, $\delta$=0.1, and $\zeta$=3 for RITUAL+VCD, and $\lambda$=0.1 and $\zeta$=3.5 for RITUAL+M3ID.

For OPERA, we set the scale factor to 50, the threshold to 15, the number of attention candidates to 5, penalty weights to 1, and the number of beams to 5.

The code is implemented in Python 3.10 with PyTorch 2.0.1 (Paszke et al., 2019), and all experiments are conducted utilizing an NVIDIA RTX 3090 GPU.

# F  ADDITIONAL EXPERIMENTS

## F.1  RANDOM IMAGE TRANSFORMATION *vs.* SINGULAR IMAGE TRANSFORMATION

In our study, we randomly choose one of six image transformation techniques (horizontal flip, vertical flip, rotate, color jitter, Gaussian blur, and crop) for the transformed image $\mathcal{V}^{(T)}$. We compared the results with a method that only adopts specific transformations rather than making a random choice.

Table 8: **Comparison of singular image transformation *vs*. random image transformation.**

| | Setup | Transformation | LLaVA 1.5 (Liu et al., 2023c) | | | |
| | | | Acc. ↑ | Prec. ↑ | Rec. ↑ | F1 ↑ |
|---|---|---|---|---|---|---|
| MS-COCO (Lin et al., 2014) | Random | Horizontal Flip | 89.50 | 89.95 | 88.93 | 89.44 |
| | | Vertical Flip | 88.60 | 88.76 | 88.40 | 88.58 |
| | | Rotate | 88.90 | 89.56 | 88.07 | 88.81 |
| | | Color Jitter | 88.83 | 89.98 | 87.40 | 88.67 |
| | | Gaussian Blur | 88.77 | 89.48 | 87.87 | 88.66 |
| | | Crop | 88.47 | 89.36 | 87.33 | 88.33 |
| | | Random Selection | 88.87 | 89.23 | 88.40 | 88.58 |
| | Popular | Horizontal Flip | 85.60 | 83.21 | 89.20 | 86.10 |
| | | Vertical Flip | 85.23 | 83.05 | 88.53 | 85.71 |
| | | Rotate | 86.20 | 84.67 | 88.40 | 86.50 |
| | | Color Jitter | 86.20 | 84.90 | 88.07 | 86.45 |
| | | Gaussian Blur | 84.93 | 83.29 | 87.40 | 85.30 |
| | | Crop | 85.70 | 84.62 | 87.27 | 85.92 |
| | | Random Selection | 85.83 | 84.17 | 88.27 | 86.17 |
| | Adversarial | Horizontal Flip | 79.50 | 74.65 | 89.33 | 81.34 |
| | | Vertical Flip | 79.10 | 74.65 | 88.13 | 80.83 |
| | | Rotate | 79.73 | 75.06 | 89.07 | 81.46 |
| | | Color Jitter | 78.70 | 74.47 | 87.33 | 80.39 |
| | | Gaussian Blur | 78.73 | 74.19 | 88.13 | 80.56 |
| | | Crop | 79.37 | 75.48 | 87.00 | 80.83 |
| | | Random Selection | 78.80 | 74.43 | 87.73 | 80.54 |

Table 9: **Effect of self-feedback on transformation selection.** While, RITUAL randomly selects image transformations, RITUAL+ selects image transformation via self-feedback from LVLMs.

| MS-COCO (Lin et al., 2014) | Method | LLaVA 1.5 (Liu et al., 2023c) | | | |
| | | Acc. ↑ | Prec. ↑ | Rec. ↑ | F1 ↑ |
|---|---|---|---|---|---|
| Random | *base* | 84.13 | 82.86 | 86.07 | 84.43 |
| | RITUAL | 88.87 | 89.23 | 88.40 | 88.81 |
| | RITUAL+ | 89.17 | 88.89 | 89.53 | 89.21 |
| Popular | *base* | 80.87 | 78.23 | 85.53 | 81.72 |
| | RITUAL | 85.83 | 84.17 | 88.27 | 86.17 |
| | RITUAL+ | 85.40 | 83.27 | 88.60 | 85.85 |
| Adversarial | *base* | 76.23 | 71.75 | 86.53 | 78.45 |
| | RITUAL | 78.80 | 74.43 | 87.73 | 80.54 |
| | RITUAL+ | 79.17 | 74.48 | 88.73 | 80.99 |

As illustrated in Table 8, our analysis revealed that the effectiveness of each augmentation varied depending on the dataset setup. For instance, employing solely color jitter led to the best results in the popular setup, while it delivered the poorest outcomes in the adversarial setup. Reviewing Figure 3, it becomes evident that the same transformation may have varying effects, beneficial or detrimental, based on the specific image and query. Therefore, we have chosen to use random selection as our primary method.

## F.2 SELF-FEEDBACK FOR TRANSFORMATION SELECTION

As we mentioned in Sec. 5 and Appendix F.1, transformation may interfere with the model's accurate predictions. To address this issue, we implemented a simple mechanism that allows the model to select an image-query-aware transformation through self-feedback. As depicted in Fig. 8, the model receives an image-question pair along with a comprehensive description of transformations, after which it selects the most suitable transformation in a self-feedback manner. Note that RITUAL+ is the model with self-feedback transformation selection rather than random choice. We compared the performance between RITUAL and RITUAL+ on POPE COCO setups in Table 9. RITUAL+ declines

**System Prompt**

A chat between a curious human and an artificial intelligence assistant. The assistant gives helpful, detailed, and polite answers to the human's questions.

------------------------------------------------------------------------------------------------

**Image**

{image sample}

------------------------------------------------------------------------------------------------

**Query**

{query sample}

------------------------------------------------------------------------------------------------

**Instruction**

You are an image augmentation evaluator. Your task is to evaluate the impact of different image augmentation techniques on question-answering task related those images. Here is the list of augmentations you need to examine:
1. Horizontal flip
- Description: Reflects the image along a vertical axis, which means that the left side of the image becomes the right side, and vice versa, while the top and bottom remain unchanged.
- Pros: Can offer a different perspective without changing the semantic meaning of the content.
- Cons: May cause issues like text becoming unreadable or objects appearing in the wrong direction.
2. Vertical flip
- Description: Flips the image along a horizontal axis, creating an upside-down version while maintaining left-right orientation.
- Pros: Useful for certain artistic effects or when orientation is not critical.
- Cons: May result in unnatural-looking images, especially if the flipped orientation affects the logic of the scene, such as objects appearing in physically impossible orientations.
3. Rotation
- Description: Alters the image orientation by a certain angle.
- Pros: Enables viewing images from different angles.
- Cons: May distort image content at extreme angles, potentially leading to the loss of important features.
4. Color jitter
- Description: Introduces variations in color, including brightness, contrast, saturation, and hue.
- Pros: Useful for simulating different lighting conditions or color variations in images.
- Cons: May introduce unrealistic colors or distortions, which can be problematic for tasks where color information is critical.
5. Gaussian blur
- Description: Applies a smoothing effect, reducing noise and fine detail.
- Pros: Helps in noise reduction and focusing on more prominent features.
- Cons: May remove important details, not suitable for tasks where fine details are crucial.
6. Crop
- Description: Removes parts of the image, focusing on a specific region of interest.
- Pros: Helps in emphasizing relevant parts of the image, potentially reducing irrelevant information.
- Cons: May remove important context or details necessary for a comprehensive understanding the image.

Consider the impact of each augmentation on the understanding of an image when answering questions. Select the most positive augmentation that helps answer questions more accurately.

Answer always in the following form:
[Number]. [Most beneficial augmentation]

For example:
1. Horizontal flip

Figure 8: **Prompt for** RITUAL+**.**

| Table 10: RITUAL with Gaussian Noise | | | | | |
|---|---|---|---|---|---|
| Noise Step | Acc. | Prec. | Rec. | F1 |
| 50 | 89.37 | 91.04 | 87.33 | 89.15 |
| 999 | 81.47 | 75.85 | 92.33 | 83.28 |

| Table 11: **VCD with Gaussian Blur** | | | | |
|---|---|---|---|---|
| Sigma | Acc. | Prec. | Rec. | F1 |
| 0.5 | 83.77 | 83.61 | 84.00 | 83.80 |
| 100 | 85.13 | 86.45 | 83.33 | 84.86 |

in the popular setting while it achieves performance improvement in random and adversarial setups. Considering the computational complexity involved in the self-feedback process, the potential for performance improvement appears limited, suggesting the need for more advanced methodologies.

### F.3 COMPARISON OF GAUSSIAN NOISE IN VCD VS GAUSSIAN BLUR IN RITUAL

We employed the standard image augmentation (e.g., crop, flip, rotate, color jitter, and Gaussian blur) techniques commonly used to enhance model robustness by generating diverse views (Chen et al., 2020; Grill et al., 2020). The key idea is that applying these augmentations at an appropriate intensity can provide diverse perspectives without compromising the underlying semantics of the image. The reason why Gaussian noise in VCD distorts the image rather than acting as a useful augmentation boils down to the intensity of the application. While we will delve into the specifics with experimental data later, the summary is that low-intensity Gaussian noise can serve as an effective augmentation. However, as the noise level increases, it shifts from providing beneficial diversity to distorting the image, which negatively impacts performance. In brief, Gaussian blur can distort the image if applied too strongly, just as Gaussian noise can serve as a diverse view generator if applied lightly. It all comes down to the intensity. Applying Gaussian noise at a low level can indeed offer a diverse perspective without compromising the image's semantics. Conversely, excessive Gaussian blur can distort the image.

To illustrate this, we conducted an experiment using Gaussian noise as a transformation within the RITUAL framework on the POPE-COCO-random setup. As shown in Table 10, Gaussian noise at low intensity (noise step=50) acts as a form of multiview augmentation, leading to positive outcomes. However, with a noise step of 999, the image became excessively distorted, impairing performance. In contrast, we also conducted VCD with Gaussian blur in Table 11. As the sigma value increases, the blur becomes stronger, leading to more significantly distorted images. In VCD, this increased distortion enhances the model's focus on the visual part of the image relative to the language part, helping to mitigate object hallucination. The stronger the image distortion, the greater the emphasis on the visual component. As a result, when the sigma value is set to 100, the distortion is more pronounced than at sigma 0.5, leading to a more substantial effect in VCD. In conclusion, both Gaussian noise and blur can provide diverse perspectives when applied moderately. However, if applied excessively, they are more likely to be perceived as distortions.

### F.4 DETAILED RESULTS ON MME-FULLSET

We present the detailed performance on MME-Fullset in Table 12. RITUAL exhibits significant performance improvement in both LLaVA-1.5 and InstructBLIP across various perception and recognition tasks in most cases. These results underscore the effectiveness of RITUAL in handling diverse tasks, including beyond the hallucination mitigation, showcasing its potential to enhance LVLMs' ability to accurately interpret and analyze visual content. However, it is important to acknowledge that RITUAL's performance in the count, position, numerical calculation, and code reasoning categories does not currently match the levels achieved in the other tasks. In the same way as shown in Fig. 6, some transformations may not suit the query and could actually contribute to a decrease in performance. Addressing and surmounting these identified drawbacks represents our primary objective moving forward.

### F.5 RESULTS OF RITUAL ON LARGER LVLMS

We report the results of the LLaVA-v1.5-13B and InstructBLIP-13B models on the POPE benchmark using the COCO dataset in Tab. 13. RITUAL achieves the best overall performance across most metrics and settings, particularly excelling in the random and popular dataset types. Although its

Table 12: **Results on MME-Fullset (Fu et al., 2024).**

| Task | Category | LLaVA 1.5 (Liu et al., 2023c) | | | | | InstructBLIP (Dai et al., 2024) | | | | |
|---|---|---|---|---|---|---|---|---|---|---|---|
| | | base | VCD | M3ID | DoLa | RITUAL | base | VCD | M3ID | DoLa | RITUAL |
| Perception | Existence | 173.75 (±4.79) | 178.75 (±2.5) | 177.50 (±5.34) | 174.58 (±5.34) | 187.50 (±2.89) | 160.42 (±5.16) | 158.75 (±7.25) | 158.33 (±5.44) | 162.08 (±5.34) | 182.50 (±6.45) |
| | Count | 121.67 (±12.47) | 126.25 (±10.4) | 124.17 (±10.93) | 122.09 (±11.73) | 139.58 (±7.62) | 79.17 (±8.22) | 90.75 (±3.11) | 94.58 (±9.85) | 82.50 (±6.16) | 74.58 (±5.99) |
| | Position | 117.92 (±3.69) | 120.00 (±4.08) | 120.00 (±7.07) | 122.09 (±2.10) | 125.00 (±10.27) | 79.58 (±8.54) | 70.00 (±15.81) | 72.50 (±17.03) | 78.75 (±8.96) | 67.08 (±10.31) |
| | Color | 149.17 (±7.51) | 150.83 (±11.01) | 152.92 (±5.67) | 149.17 (±4.19) | 164.17 (±6.87) | 130.42 (±17.34) | 132.5 (±18.78) | 128.33 (±14.72) | 135.42 (±10.49) | 139.17 (±0.96) |
| | Posters | 124.24 (±3.36) | 129.34 (±4.11) | 120.49 (±8.23) | 127.98 (±5.51) | 135.46 (±0.94) | 101.96 (±1.5) | 114.29 (±7.07) | 110.54 (±0.62) | 105.10 (±3.41) | 139.46 (±4.85) |
| | Celebrity | 115.44 (±3.98) | 124.78 (±6.23) | 113.9 (±4.85) | 115.00 (±8.20) | 120.07 (±1.88) | 105.22 (±2.23) | 128.31 (±5.14) | 119.05 (±5.01) | 150.74 (±2.15) | 134.63 (±4.19) |
| | Scene | 147.44 (±6.26) | 152.69 (±2.46) | 155.94 (±2.83) | 150.94 (±1.21) | 159.75 (±2.79) | 130.19 (±3.9) | 140.56 (±2.92) | 145.31 (±5.78) | 147.75 (±4.98) | 158.63 (±2.62) |
| | Landmark | 133.31 (±4.73) | 136.00 (±7.35) | 133.81 (±5.84) | 132.31 (±6.20) | 157.81 (±2.19) | 118.13 (±6.37) | 131.06 (±3.71) | 127.06 (±7.17) | 126.31 (±3.68) | 150.69 (±1.39) |
| | Artwork | 107.31 (±2.61) | 110.50 (±0.79) | 111.69 (±0.92) | 107.25 (±7.95) | 117.31 (±2.23) | 91.44 (±5.61) | 102.75 (±4.24) | 98.44 (±3.91) | 117.44 (±4.31) | 103.94 (±6.95) |
| | OCR | 107.50 (±13.99) | 98.13 (±7.18) | 112.50 (±10.21) | 97.50 (±10.80) | 121.25 (±6.29) | 90.63 (±6.88) | 81.25 (±6.61) | 78.75 (±17.85) | 73.13 (±8.00) | 93.75 (±8.29) |
| Recognition | Commonsense Reasoning | 99.82 (±9.39) | 108.04 (±2.36) | 107.32 (±10.13) | 107.32 (±8.98) | 115.54 (±4.92) | 92.68 (±8.64) | 92.86 (±6.20) | 96.43 (±9.70) | 96.43 (±1.31) | 109.11 (±8.17) |
| | Numerical Calculation | 60.00 (±12.42) | 63.75 (±8.54) | 68.75 (±7.22) | 64.38 (±12.64) | 52.50 (±8.9) | 56.88 (±15.6) | 64.38 (±6.25) | 60.63 (±19.51) | 56.88 (±11.97) | 63.75 (±9.24) |
| | Text Translation | 81.88 (±13.13) | 77.50 (±8.90) | 87.50 (±10.61) | 81.25 (±8.78) | 93.75 (±10.51) | 56.88 (±17.49) | 66.25 (±6.61) | 72.50 (±12.75) | 74.38 (±10.48) | 89.38 (±12.48) |
| | Code Reasoning | 64.38 (±25.93) | 63.75 (±25.86) | 64.38 (±25.93) | 64.38 (±29.04) | 65.00 (±10.21) | 63.75 (±11.27) | 72.50 (±20.31) | 78.13 (±15.33) | 70.00 (±7.91) | 70.00 (±4.08) |

Table 13: **Results of 13B models on COCO dataset.**

| Setup | Method | LLaVA-1.5 (13B) | | | | InstructBLIP (13B) | | | |
|---|---|---|---|---|---|---|---|---|---|
| | | Acc. | Prec. | Rec. | F1 | Acc. | Prec. | Rec. | F1 |
| Random | *base* | 82.70 | 78.73 | 89.60 | 83.82 | 80.10 | 75.21 | 89.80 | 81.86 |
| | VCD | 82.97 | 79.00 | 89.80 | 84.06 | 82.83 | 78.65 | 90.13 | 84.00 |
| | M3ID | 84.53 | 80.51 | 91.13 | 85.49 | 81.57 | 76.56 | 91.00 | 83.16 |
| | RITUAL | 87.03 | 83.69 | 92.00 | 87.65 | 84.87 | 78.49 | 96.07 | 86.39 |
| Popular | *base* | 80.93 | 76.95 | 88.33 | 82.25 | 75.80 | 70.14 | 89.87 | 78.78 |
| | VCD | 80.23 | 75.58 | 89.33 | 81.88 | 77.43 | 71.56 | 91.07 | 80.14 |
| | M3ID | 81.57 | 76.92 | 90.20 | 83.03 | 76.43 | 70.22 | 91.80 | 79.57 |
| | RITUAL | 84.57 | 80.20 | 91.80 | 85.61 | 78.43 | 71.23 | 95.40 | 81.56 |
| Adversarial | *base* | 75.90 | 70.76 | 88.27 | 78.55 | 71.47 | 65.48 | 90.80 | 76.09 |
| | VCD | 75.63 | 69.83 | 90.27 | 78.74 | 73.33 | 67.45 | 90.20 | 77.18 |
| | M3ID | 78.77 | 73.09 | 91.07 | 81.09 | 71.40 | 65.29 | 91.40 | 76.17 |
| | RITUAL | 77.93 | 71.75 | 92.13 | 80.68 | 72.37 | 65.37 | 95.13 | 77.49 |

performance slightly falls short of VCD and M3ID under the adversarial setting, its superiority in other types suggests its robustness and effectiveness.

Moreover, we extend our experiments to the additional larger LVLM, mPLUG-owl2 (Ye et al., 2024). As shown in Table 14. our proposed RITUAL demonstrates the best performance in most cases on the POPE benchmark, similar to its success with LLaVa and InstructBLIP. This highlights the versatility and robustness of our approach across different LVLMs.

### F.6 RESULTS OF MULTIPLE AUGMENTED IMAGES OF RITUAL

As shown in Tab. 15, we found that performances slightly improve with the addition of more augmented images. This improvement is likely due to the increased variety of views available for the same scene, enhancing the model's generalization ability. However, it is important to note that this also leads to increased computational overhead due to the necessity of additional forward passes.

Table 14: **Results of mPLUG-owl2 (Ye et al., 2024) on POPE benchmark.**

| Dataset | Setup | Method | mPLUG-owl2 | | | |
|---|---|---|---|---|---|---|
| | | | Acc. | Prec. | Rec. | F1 |
| COCO | Random | *base* | 81.00 | 75.27 | 92.33 | 82.93 |
| | | VCD | 81.53 | 76.40 | 91.27 | 83.17 |
| | | M3ID | 80.90 | 75.29 | 92.00 | 82.81 |
| | | DoLa | 81.20 | 75.97 | 91.27 | 82.92 |
| | | RITUAL | 84.83 | 80.40 | 92.13 | 85.87 |
| | Popular | *base* | 76.27 | 69.96 | 92.07 | 79.50 |
| | | VCD | 75.70 | 69.88 | 90.33 | 78.80 |
| | | M3ID | 76.50 | 70.23 | 92.00 | 79.65 |
| | | DoLa | 76.67 | 70.58 | 91.47 | 79.67 |
| | | RITUAL | 80.43 | 74.64 | 92.20 | 82.49 |
| | Adversarial | *base* | 73.20 | 66.88 | 91.93 | 77.43 |
| | | VCD | 73.23 | 67.26 | 90.53 | 77.18 |
| | | M3ID | 72.57 | 66.28 | 91.87 | 77.00 |
| | | DoLa | 72.37 | 66.29 | 91.00 | 76.71 |
| | | RITUAL | 75.23 | 68.88 | 92.07 | 78.80 |
| A-OKVQA | Random | *base* | 78.13 | 70.87 | 95.53 | 81.37 |
| | | VCD | 77.70 | 70.42 | 95.53 | 81.07 |
| | | M3ID | 78.23 | 70.73 | 96.33 | 81.57 |
| | | DoLa | 77.67 | 70.38 | 95.53 | 81.05 |
| | | RITUAL | 80.20 | 73.02 | 95.80 | 82.87 |
| | Popular | *base* | 71.27 | 64.43 | 94.93 | 76.77 |
| | | VCD | 71.07 | 64.21 | 95.20 | 76.69 |
| | | M3ID | 69.57 | 62.80 | 96.00 | 75.93 |
| | | DoLa | 71.10 | 64.22 | 95.27 | 76.72 |
| | | RITUAL | 74.20 | 66.96 | 95.53 | 78.74 |
| | Adversarial | *base* | 64.83 | 59.15 | 95.87 | 73.16 |
| | | VCD | 66.43 | 60.39 | 95.53 | 74.00 |
| | | M3ID | 65.13 | 59.33 | 96.27 | 73.41 |
| | | DoLa | 65.73 | 59.91 | 95.13 | 73.52 |
| | | RITUAL | 65.93 | 59.99 | 95.67 | 73.74 |
| GQA | Random | *base* | 80.00 | 74.04 | 92.40 | 82.21 |
| | | VCD | 81.60 | 77.56 | 88.93 | 82.86 |
| | | M3ID | 80.93 | 74.95 | 92.93 | 82.98 |
| | | DoLa | 78.67 | 73.19 | 90.47 | 80.92 |
| | | RITUAL | 82.10 | 76.10 | 93.60 | 83.95 |
| | Popular | *base* | 71.53 | 64.94 | 93.60 | 76.68 |
| | | VCD | 71.40 | 65.77 | 89.27 | 75.74 |
| | | M3ID | 71.50 | 65.06 | 92.87 | 76.52 |
| | | DoLa | 71.03 | 65.23 | 90.07 | 75.67 |
| | | RITUAL | 73.47 | 66.60 | 94.13 | 78.01 |
| | Adversarial | *base* | 68.73 | 62.60 | 93.07 | 74.85 |
| | | VCD | 71.67 | 65.98 | 89.47 | 75.95 |
| | | M3ID | 68.23 | 62.29 | 92.40 | 74.42 |
| | | DoLa | 69.50 | 63.51 | 91.67 | 75.03 |
| | | RITUAL | 68.30 | 62.15 | 93.60 | 74.70 |

Table 15: **Ablation of the number of augmented images in** RITUAL **on COCO dataset.**

| Setup | # of Aug. Images | LLaVA-1.5 | | | |
|---|---|---|---|---|---|
| | | Acc. | Prec. | Rec. | F1 |
| Random | 1 | 88.87 | 89.23 | 88.40 | 88.81 |
| | 2 | 89.07 | 89.38 | 88.67 | 89.02 |
| | 3 | 89.17 | 89.25 | 89.07 | 89.16 |
| Popular | 1 | 85.83 | 84.17 | 88.27 | 86.17 |
| | 2 | 85.37 | 83.85 | 87.60 | 85.69 |
| | 3 | 86.20 | 84.11 | 89.27 | 86.61 |
| Adversarial | 1 | 78.80 | 74.43 | 87.73 | 80.54 |
| | 2 | 79.10 | 74.56 | 88.33 | 80.87 |
| | 3 | 79.07 | 74.63 | 88.07 | 80.80 |

Table 16: **Comparison of yes ratio with respect to the additional query "Please answer this question with one word." of LLaVA 1.5 on COCO random setup.**

| Additional Query | Method | Yes Ratio | Acc. | Prec. | Rec. | F1 |
|---|---|---|---|---|---|---|
| ✓ | *base* | 39.90 | 83.29 | 92.13 | 72.80 | 81.33 |
| | VCD | 40.97 | 87.73 | 91.42 | 83.28 | 87.16 |
| | *base* | 51.87 | 84.13 | 82.86 | 86.07 | 84.43 |
| | VCD | 53.37 | 85.37 | 83.14 | 88.73 | 85.84 |
| | M3ID | 50.97 | 86.00 | 85.11 | 87.27 | 86.18 |
| | DoLa | 51.23 | 85.97 | 85.10 | 87.20 | 86.14 |
| | RITUAL | 49.53 | 88.87 | 89.23 | 88.40 | 88.81 |

Using multiple augmented images can indeed contribute to performance improvement, but it comes with the inherent trade-off of increased latency due to the additional computational cost.

### F.7 IMPACT OF ONE WORD CONSTRAINT

The VCD setup prompts the model with an additional instruction, "Please answer this question with one word," at the end of each question. As shown in Tab. 16, this constraint biases the model towards shorter, more definitive answers, with a notable inclination towards "No" (with a No ratio of 60In contrast, our evaluation setup does not include this "one word" constraint. Instead, we allow the model to generate more detailed responses that include explanations. This approach tends to yield a balanced "Yes" and "No" ratio. Consequently, our method evaluates whether the generated output contains a "Yes" or "No" along with the explanation, rather than restricting the output to a single word for simplicity in evaluation. To provide more context, we have included a Tab. 16 that presents the performance metrics under different settings with the respective "Yes" ratios. By removing the "one word" constraint, we aim to capture more nuanced and contextually rich responses from the model, which we believe provides a more comprehensive assessment of its capabilities. Additionally, since there is no official implementation of M3ID, we reimplemented it and reported the results based on our settings.

### F.8 EFFECT OF $\alpha$ IN RITUAL

As shown in Table 17, we conduct an ablation study on the hyperparameter $\alpha$ in Eq. (4), which adjusts the ratio between the output logits of the model conditioned on the original image $\mathcal{V}$ and the transformed image $\mathcal{V}^{(\mathcal{T})}$. We vary $\alpha$ from 0 (standard decoding) to 3.5 on the POPE COCO random setting. Our method consistently outperforms the baseline across a broad spectrum of $\alpha$ values, with accuracy improvement ranging from $+3.60$ to $+4.74$. This demonstrates that our approach is robust and effective regardless of the specific hyperparameter value chosen. Based on these results, we set $\alpha = 3$ as the default value.

Table 17: **Ablation of $\alpha$ on POPE (Li et al., 2023c) COCO random.** Based on the results, we set $\alpha = 3$ as the default.

| $\alpha$ | Acc. ↑ | Prec. ↑ | Rec. ↑ | F1 ↑ |
|---|---|---|---|---|
| 0 (*base*) | 84.13 | 82.86 | 86.07 | 84.43 |
| 0.5 | 87.73 | 87.04 | 88.67 | 87.85 |
| 1 | 88.00 | 87.70 | 88.40 | 88.05 |
| 1.5 | 88.53 | 88.74 | 88.27 | 88.50 |
| 2 | 88.50 | 89.05 | 87.80 | 88.42 |
| 2.5 | 88.27 | 88.68 | 87.73 | 88.20 |
| 3 | 88.87 | 89.23 | 88.40 | 88.81 |
| 3.5 | 88.67 | 89.40 | 87.73 | 88.56 |

Table 18: **Confusion matrices on POPE (Li et al., 2023c) benchmark.**

| | Setup | Method | LLaVA 1.5 (Liu et al., 2023c) | | | | | InstructBLIP (Dai et al., 2024) | | | | |
|---|---|---|---|---|---|---|---|---|---|---|---|---|
| | | | TP ↑ | FP ↓ | TN ↑ | FN ↓ | Acc. ↑ | TP ↑ | FP ↓ | TN ↑ | FN ↓ | Acc. ↑ |
| MS-COCO (Lin et al., 2014) | Random | *base* | 1291 | 267 | 1233 | 209 | 84.13 | 1255 | 271 | 1229 | 245 | 82.80 |
| | | VCD | 1331 | 270 | 1230 | 169 | 85.37 | 1240 | 222 | 1278 | 260 | 83.93 |
| | | M3ID | 1309 | 229 | 1271 | 191 | 86.00 | 1260 | 229 | 1271 | 240 | 84.37 |
| | | RITUAL | 1326 | 160 | 1340 | 174 | 88.87 | 1302 | 137 | 1363 | 198 | 88.83 |
| | | RITUAL+VCD | 1323 | 154 | 1346 | 177 | 89.07 | 1311 | 132 | 1368 | 189 | 89.30 |
| | | RITUAL+M3ID | 1319 | 149 | 1351 | 181 | 89.00 | 1294 | 126 | 1374 | 206 | 88.93 |
| | Popular | *base* | 1283 | 357 | 1143 | 217 | 80.87 | 1238 | 464 | 1036 | 262 | 75.80 |
| | | VCD | 1306 | 373 | 1127 | 194 | 81.10 | 1234 | 402 | 1098 | 266 | 77.73 |
| | | M3ID | 1324 | 339 | 1161 | 176 | 82.83 | 1259 | 440 | 1060 | 241 | 77.30 |
| | | **Ours** | 1324 | 249 | 1251 | 176 | 85.83 | 1309 | 350 | 1150 | 191 | 81.97 |
| | | RITUAL+VCD | 1328 | 255 | 1245 | 172 | 85.77 | 1309 | 324 | 1176 | 191 | 82.83 |
| | | RITUAL+M3ID | 1320 | 259 | 1241 | 180 | 85.37 | 1304 | 347 | 1153 | 196 | 81.90 |
| | Adversarial | *base* | 1298 | 511 | 989 | 202 | 76.23 | 1263 | 501 | 999 | 237 | 75.40 |
| | | VCD | 1308 | 540 | 960 | 192 | 75.60 | 1253 | 449 | 1051 | 247 | 76.80 |
| | | M3ID | 1310 | 479 | 1021 | 190 | 77.70 | 1259 | 478 | 1022 | 241 | 76.03 |
| | | RITUAL | 1316 | 452 | 1048 | 184 | 78.80 | 1308 | 446 | 1054 | 192 | 78.73 |
| | | RITUAL+VCD | 1323 | 435 | 1065 | 177 | 79.60 | 1312 | 440 | 1060 | 188 | 79.07 |
| | | RITUAL+M3ID | 1320 | 444 | 1056 | 180 | 79.20 | 1300 | 432 | 1068 | 200 | 78.93 |
| A-OKVQA (Schwenk et al., 2022) | Random | *base* | 1373 | 421 | 1079 | 127 | 81.73 | 1300 | 366 | 1134 | 200 | 81.13 |
| | | VCD | 1405 | 450 | 1050 | 95 | 81.83 | 1297 | 337 | 1163 | 203 | 82.00 |
| | | M3ID | 1407 | 400 | 1100 | 93 | 83.57 | 1357 | 387 | 1113 | 143 | 82.33 |
| | | RITUAL | 1413 | 358 | 1142 | 87 | 85.17 | 1378 | 264 | 1236 | 122 | 87.13 |
| | | RITUAL+VCD | 1406 | 353 | 1147 | 94 | 85.10 | 1373 | 270 | 1230 | 127 | 86.77 |
| | | RITUAL+M3ID | 1419 | 341 | 1159 | 81 | 85.93 | 1369 | 254 | 1246 | 131 | 87.17 |
| | Popular | *base* | 1375 | 575 | 925 | 125 | 76.67 | 1303 | 533 | 967 | 197 | 75.67 |
| | | VCD | 1393 | 652 | 848 | 107 | 74.70 | 1314 | 519 | 981 | 186 | 76.50 |
| | | M3ID | 1416 | 551 | 949 | 84 | 78.83 | 1375 | 513 | 987 | 125 | 78.73 |
| | | RITUAL | 1416 | 551 | 949 | 84 | 78.83 | 1375 | 513 | 987 | 125 | 78.73 |
| | | RITUAL+VCD | 1414 | 539 | 961 | 86 | 79.17 | 1383 | 518 | 982 | 117 | 78.83 |
| | | RITUAL+M3ID | 1418 | 529 | 971 | 82 | 79.63 | 1373 | 497 | 1003 | 127 | 79.20 |
| | Adversarial | *base* | 1369 | 847 | 653 | 131 | 67.40 | 1302 | 762 | 738 | 198 | 68.00 |
| | | VCD | 1400 | 877 | 623 | 100 | 67.43 | 1327 | 707 | 793 | 173 | 70.67 |
| | | M3ID | 1404 | 861 | 639 | 96 | 68.10 | 1326 | 739 | 761 | 174 | 69.57 |
| | | RITUAL | 1414 | 857 | 643 | 86 | 68.57 | 1378 | 770 | 730 | 122 | 70.27 |
| | | RITUAL+VCD | 1412 | 848 | 652 | 88 | 68.80 | 1385 | 755 | 745 | 115 | 71.00 |
| | | RITUAL+M3ID | 1415 | 852 | 648 | 85 | 68.77 | 1367 | 788 | 712 | 133 | 69.30 |
| GQA (Hudson & Manning, 2019) | Random | *base* | 1390 | 453 | 1047 | 110 | 81.23 | 1289 | 391 | 1109 | 211 | 79.93 |
| | | VCD | 1426 | 481 | 1019 | 74 | 81.50 | 1300 | 345 | 1155 | 200 | 81.83 |
| | | M3ID | 1417 | 432 | 1068 | 83 | 82.83 | 1315 | 398 | 1102 | 185 | 80.57 |
| | | RITUAL | 1435 | 352 | 1148 | 65 | 86.10 | 1327 | 281 | 1219 | 173 | 84.87 |
| | | RITUAL+VCD | 1435 | 354 | 1146 | 65 | 86.03 | 1334 | 285 | 1215 | 166 | 84.97 |
| | | RITUAL+M3ID | 1433 | 344 | 1156 | 67 | 86.30 | 1322 | 272 | 1228 | 178 | 85.00 |
| | Popular | *base* | 1402 | 727 | 773 | 98 | 72.50 | 1281 | 599 | 901 | 219 | 72.73 |
| | | VCD | 1422 | 775 | 725 | 78 | 71.57 | 1298 | 588 | 912 | 202 | 73.67 |
| | | M3ID | 1410 | 725 | 775 | 90 | 72.83 | 1316 | 579 | 921 | 184 | 74.57 |
| | | RITUAL | 1435 | 691 | 809 | 65 | 74.80 | 1326 | 591 | 909 | 174 | 74.50 |
| | | RITUAL+VCD | 1431 | 679 | 821 | 69 | 75.07 | 1331 | 571 | 929 | 169 | 75.33 |
| | | RITUAL+M3ID | 1433 | 701 | 799 | 67 | 74.40 | 1331 | 564 | 936 | 169 | 75.57 |
| | Adversarial | *base* | 1397 | 868 | 632 | 103 | 67.63 | 1285 | 698 | 802 | 215 | 69.57 |
| | | VCD | 1413 | 889 | 611 | 87 | 67.47 | 1279 | 696 | 804 | 221 | 69.43 |
| | | M3ID | 1417 | 873 | 627 | 83 | 68.13 | 1292 | 725 | 775 | 208 | 68.90 |
| | | RITUAL | 1437 | 890 | 610 | 63 | 68.23 | 1327 | 722 | 778 | 173 | 70.17 |
| | | RITUAL+VCD | 1435 | 865 | 635 | 65 | 69.00 | 1328 | 721 | 779 | 172 | 70.23 |
| | | RITUAL+M3ID | 1429 | 865 | 635 | 71 | 68.80 | 1343 | 713 | 787 | 157 | 71.00 |

## F.9 CONFUSION MATRICES OF LLaVA-1.5

To analyze the performance of the model in detail, we report the confusion matrices in Table 18 for the POPE benchmark. Notably, RITUAL significantly improves True Negatives (TN) while maintaining a

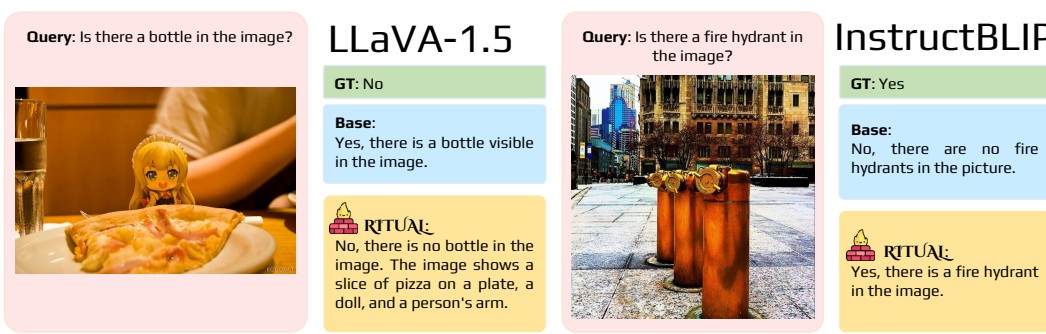

Figure 9: **Results on POPE (Li et al., 2023c).**

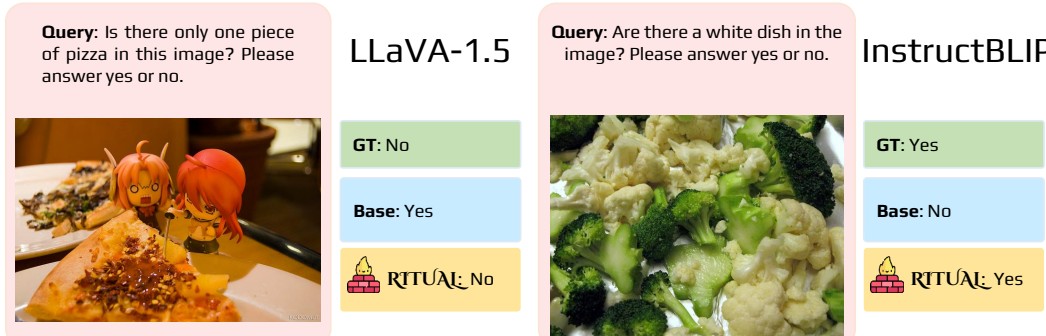

Figure 10: **Results on MME (Fu et al., 2024).**

similar level of True Positives (TP) compared to existing contrastive decoding methods. It implies that our method achieves the highest accuracy by significantly improving the identification of non-relevant instances compared to the baseline and previous methods.

### F.10 QUALITATIVE EXAMPLES

We provide qualitative examples on all benchmarks (POPE (Li et al., 2023c), MME (Fu et al., 2024), CHAIR (Rohrbach et al., 2018), and LLaVA-Bench (Liu et al., 2023c)) to verify the efficacy of RITUAL. Results are shown in Figs. 9 to 12.

## G LICENSE OF ASSETS

POPE (Li et al., 2023c) is licensed under MIT License. CHAIR (Rohrbach et al., 2018) is made available under the BSD 2-Clause License. LLaVA-Bench is available under Apache-2.0 License.

LLaVA (Liu et al., 2023c) is licensed under the Apache-2.0 License. InstructBLIP (Dai et al., 2024) is under BSD-3-Clause License.

PyTorch (Paszke et al., 2019) is released under the Modified BSD License.

## H LIMITATIONS

- **Computational overhead**: RITUAL requires running the model two times for each test image, which increases inference time and computational load. This can be problematic in real-time or resource-constrained environments.

- **Diminishing returns**: While RITUAL can provide performance boosts, there are diminishing returns. Excessive augmentation or unnecessary augmentations may not yield significant improvements and could add unnecessary complexity.

**Query**: Please describe this image in details.

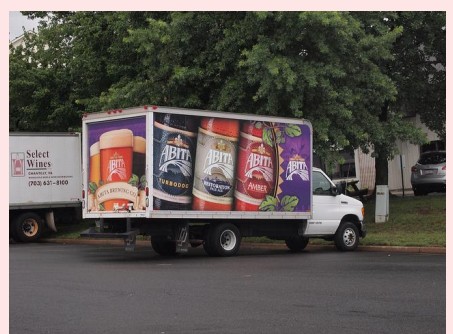

# LLaVA-1.5

**Base**:
The image depicts a large white moving truck, presumably carrying a delivery of brown bottles, as it either stops or drives down a road. The truck takes up a significant part of the scene, covering more than half of the image. It is parked near the side of a building, with ...

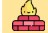 **RITUAL**:
The image features a white truck parked on a street, with a large advertisement for beer displayed on the back of the truck. The truck is positioned near a tree, and there is another truck visible in the background. The advertisement on the truck showcases a variety ...

**Query**: Please describe this image in details.

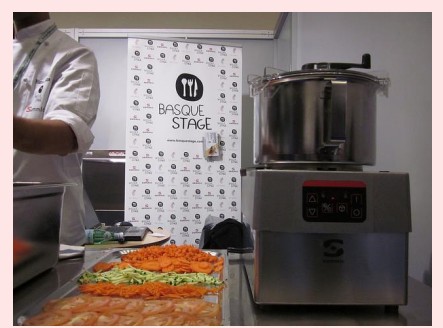

# InstructBLIP

**Base**:
The image displays a busy stage at a restaurant or food preparation facility with several chefs or cooks preparing meals. There are nine people present in the room. In addition to the cooks, there is a person observing the work from behind a fence, likely assessing the progress of the ...

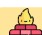 **RITUAL**:
The image features a kitchen with a man preparing food in a professional setting. He is standing in front of a counter with a large food processor, which is positioned towards the left side of the scene. The man is wearing a chef's hat and is focused on preparing the food. There ...

Figure 11: **Results on CHAIR (Rohrbach et al., 2018).**

- **Task-specific augmentations**: The choice of augmentations needs to be task-specific. For instance, while horizontal flips might be useful in natural image classification, they may not be appropriate for medical imaging or charts (*e.g.*, flipping a chest X-ray might create unrealistic artifacts).

- **May not always effective for all models**: Some models, especially highly regularized or robust models, may not benefit as much from RITUAL if they are already trained on extensive data augmentations during training.

RITUAL is a powerful technique that improves model robustness, particularly in scenarios where test data is ambiguous. It leverages transformations of the input data to achieve better generalization. However, it comes with trade-offs in terms of increased inference time and computational load, which should be balanced against the expected performance gains.

# I    BROADER IMPACTS

The broader impacts of proposed RITUAL have benefits and risks along with its release.

[+] **Increased Reliability in Critical Applications.** By mitigating hallucinations in LVLMs, we can significantly enhance the reliability of these models in critical applications such as medical diagnosis, autonomous driving, and surveillance. This leads to more accurate and dependable outcomes, which are crucial for safety and effectiveness in these fields.

**Query**: How many uncut fruits are in the image?

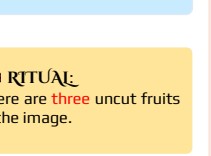

**Base**:
There are four uncut fruits in the image.

🔥 RITUAL:
There are three uncut fruits in the image.

**Query**: How many coffee mugs are in the set?

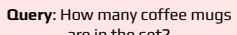
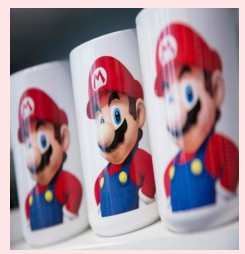

**Base**:
There are four coffee mugs in the set.

🔥 RITUAL:
There are three coffee mugs in the set.

**Query**: This is the logo of LLaVA, Large Language and Vision Assistant, based on the LLaMA architecture. Please explain this logo in detail, and how do you think of its design.

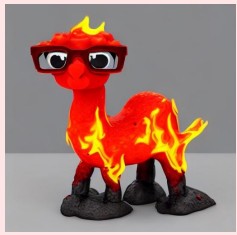

**Base**:
Sure, I'd be happy to explain the concept and design behind the LLaVA logo. Based on the image, the LLaVA logo features a fire-breathing red flaming luchador, which may represent a mix of courage, creative expression, and energy.

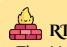

🔥 RITUAL:
The LLaVA logo features a unique design, combining the shape of a horse with the flames of a fire. The horse is depicted in a red color, while the flames are shown in orange. The combination of these two elements creates an eye-catching and memorable visual representation of the brand.

**Query**: Can you explain this meme in detail?

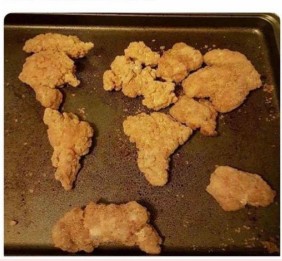

**Base**:
This meme is a composite image featuring a fried-chicken centerpiece sitting on a pan, with the shape of the dish taking the form of a map. The large piece of fried chicken consists of various pieces, distributed all over the image. A quote, presumably from Garfield, is included at the top of the image, stating, \"The earth is just spinning by and I'm frying chicken at home. How beautiful it all is.\" This amusing combination of elements defines a unique and intriguing aspect of food presentation in a visually appealing manner.

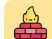

🔥 RITUAL:
The image features a tray of fried chicken pieces arranged to resemble a map of the world. The chicken pieces are spread across the tray, with some pieces placed near the edges and others in the center. The chicken pieces are shaped like continents and countries, creating a unique and creative representation of the world.

Figure 12: **Additional case studies of LLaVA-1.5 on LLaVA-Bench (Liu et al., 2023c).** Hallucinations are highlighted in red.

[+] **Reduction in Misinformation.** Reducing hallucinations helps minimize the spread of misinformation in applications like news generation or content moderation, thereby contributing to more accurate and trustworthy information dissemination.

[−] **Increased Computational Costs.** Implementing our hallucination mitigation technique RITUAL requires two times forward passes, which can lead to increased costs and energy consumption.

Despite the potential negative impacts, the positive aspects of RITUAL far outweigh the drawbacks. Enhancing trustworthiness in LVLMs is a crucial issue, and we hope our work stimulates the research community to develop more effective hallucination mitigation strategies to address it.

