# OpenReview forum: "RITUAL: Random Image Transformations as a Universal Anti-hallucination Lever in LVLMs"
_ICLR.cc/2025/Conference — ICLR 2025 Conference Withdrawn Submission_

### Official Review · Reviewer_pTiT · 2024-10-19

**Soundness:** 3
**Presentation:** 3
**Contribution:** 2
**Rating:** 5
**Confidence:** 5

**Summary:**

This paper presents RITUAL, a decoding method to reduce hallucinations in Large Vision Language Models (LVLMs). Inspired by test-time augmentation (TTA), RITUAL applies random image transformations during decoding to diversify inputs, improving the model's robustness. By combining transformed images with the original input, the method adjusts probability distributions to correct hallucination in LVLMs.

**Strengths:**

* **Clarity and simplicity**: The proposed method is straightforward and easy to implement.
* **Demonstrated effectiveness and versatility**: The experimental results highlight the method's effectiveness. As a decoding approach for LVLM, RITUAL exhibits general applicability, showing improvement not only in object hallucination reduction but also across some general-purpose benchmarks.

**Weaknesses:**

* **Limited novelty**: The contribution is somewhat incremental. The method follows visual contrastive decoding (VCD) [1], with a modification where random transformations and distribution aggregation are employed in place of VCD’s noise injection and distribution subtraction. While the modification is effective, it primarily reflects an engineering improvement (I have read Appendix B and F3).
* **Concerns about image transformation**: In the main paper, the authors propose that random image transformations help calibrate the output distribution, but I have concerns about its impact on image semantics. For example, perturbations such as changes in **color jitter or flipping** might alter the semantic content of the image, leading to inconsistencies in model predictions (e.g., in tasks such as **image captioning** and **VQA**). This could introduce conflicts during decoding.
* **Self-feedback for transformation selection**: I am confused that why the authors introduced an transformation selection strategy **in the appendix** to address the aforementioned concern. Moreover, though this is a reasonable solution, it increases the computational burden, adding further inference burden to an already expensive decoding process.
* **Limited evaluation**: The evaluation of caption hallucinations relies solely on the CHAIR metric, which takes a relatively simplistic approach by parsing all object properties in the sentence. I recommend the authors incorporate LLM-based evaluations like SHR [2] for a more nuanced assessment. Additionally, integrating advanced hallucination benchmarks, such as HalluBench [3], would strengthen the validity of the proposed method.

[1] Leng, Sicong, et al. "Mitigating object hallucinations in large vision-language models through visual contrastive decoding." *Proceedings of the IEEE/CVF Conference on Computer Vision and Pattern Recognition*. 2024.

[2] Zhao, Zhiyuan, et al. "Beyond hallucinations: Enhancing lvlms through hallucination-aware direct preference optimization." *arXiv preprint https://arxiv.org/abs/2311.16839* (2023).

[3] Guan, Tianrui, et al. "HallusionBench: an advanced diagnostic suite for entangled language hallucination and visual illusion in large vision-language models." *Proceedings of the IEEE/CVF Conference on Computer Vision and Pattern Recognition*. 2024.

**Questions:**

All relevant questions are included under the “Weaknesses” section.

---

### Official Review · Reviewer_4JcU · 2024-11-02

**Soundness:** 2
**Presentation:** 3
**Contribution:** 2
**Rating:** 5
**Confidence:** 3

**Summary:**

The paper uses test-time augmentation to alleviate the hallucination of Vlms. The VLM response is sampled from a linear combination of the two probability distributions based on the two visual inputs.

**Strengths:**

1) The paper and the method are easy to follow and clear.
2) The method is simple and easy to use.

**Weaknesses:**

1) How does RITUAL differ from or improve upon the test-time augmentation methods described in Shanmugam et al. (2020)? It would be helpful if the authors could provide a more detailed comparison with this and other relevant prior work on test-time augmentation.
[1] Shanmugam, Divya, et al. "When and why test-time augmentation works." arXiv preprint arXiv:2011.11156 1.3 (2020): 4.

Please refer to the questions.

**Questions:**

1) Have the authors explored alternative methods for combining the two distributions, such as the aggregation techniques proposed by Shanmugam et al. (2021)? If so, how do these compare to the linear combination approach? If not, could the authors discuss why they chose linear combination and whether they believe other aggregation methods might be beneficial?.
Shanmugam, Divya, et al. "Better aggregation in test-time augmentation." Proceedings of the IEEE/CVF international conference on computer vision. 2021.

2) The paper says the contrastive methods need to forward twice and RITUAL does not. However, from my understanding, RITUAL also needs to forward twice.

3) The paper states that the OPERA's results are high due to it's using beam search. Can RITUAL also use beam search? What are the results then?

---

### Official Review · Reviewer_XhiP · 2024-11-03

**Soundness:** 2
**Presentation:** 3
**Contribution:** 2
**Rating:** 5
**Confidence:** 4

**Summary:**

RITUAL is a training-free method to improve robustness against hallucinations in Large Vision Language Models (LVLMs) by introducing random image transformations as complementary inputs during decoding. This technique reduces hallucinations by exposing the model to diverse visual scenarios, aiding in more accurate decision-making without requiring additional models or complex feedback mechanisms. RITUAL outperforms existing methods in object hallucination benchmarks, offering a straightforward and effective solution.

**Strengths:**

1. **Simplicity and Practicality:** RITUAL is a straightforward, training-free method without requiring additional models, making it easy to implement and deploy.

2. **Effective in Reducing Hallucinations:** By introducing random image transformations during decoding, RITUAL effectively reduces hallucinations, as evidenced by significant improvements across multiple benchmarks like POPE, CHAIR, and MME.

3. **High Transferability:** Since RITUAL doesn't involve modifications to the LVLM training process, it is easily adaptable to a variety of models and compatible with existing contrastive decoding methods.

4. **Well-Written and Accessible Paper:** The clear and concise writing in this paper enhances understanding, making the proposed method and its results accessible to a wide audience.

**Weaknesses:**

1. The experimental results are not fully consistent across different models, sometimes showing improvement and sometimes underperforming. This variability, possibly due to the randomness of data augmentations, raises concerns about the method's reliability. Although the authors have introduced RITUAL+ to solve this problem and provided some analysis, more robust testing is needed to confirm consistent effectiveness.

2. The experiments are conducted on a small selection of models, namely LLaVA, InstructBLIP, and mPLUG-OWL2. To demonstrate broader applicability, testing on more powerful models like GPT-4V or other LVLMs would be beneficial.

3. The method's performance in real-world applications may differ from benchmark scenarios. Since RITUAL does not guarantee a high likelihood of improvement in every inference instance, its value is somewhat limited for practical deployment, where reliability is essential.

**Questions:**

1. **How does RITUAL perform on more advanced LVLMs like GPT-4V?** Since only a few models were tested, it would be valuable to understand whether this approach holds up or possibly improves on other models.

2. **Could additional experiments validate the impact of RITUAL+ on reducing hallucinations?** While RITUAL+ was introduced as an enhancement, further testing could clarify its benefits and any possible limitations compared to the original RITUAL method.

---

### Official Review · Reviewer_voyT · 2024-11-03

**Soundness:** 2
**Presentation:** 3
**Contribution:** 2
**Rating:** 5
**Confidence:** 4

**Summary:**

This paper presents a simple and efficient method to mitigates hallucinations by just  using the original and transformed images in the inference phase. Experimental results demonstrated that RITUAL  significantly outperforms existing contrastive decoding methods across several object hallucination benchmarks, including POPE, CHAIR, and MME.

**Strengths:**

1）The method proposed in this paper reduces hallucination issues in LVLMs through random
image transformations. It significantly improves the model's performance across multiple
benchmarks without requiring additional training or complex external mechanisms.
2）This method is compatible with existing contrastive decoding approaches and can improve
performance without relying on additional models, demonstrating excellent adaptability
across different models and tasks.

**Weaknesses:**

1）The method lacks significant innovation, primarily relying on the combination of image
enhancement and contrastive decoding techniques.
2. The lack of in-depth analysis on the performance of each transformation in different tasks
may result in the method being unstable in practical applications. Some transformations
may be beneficial in specific tasks but counterproductive in others. This arbitrary selection of
transformations lacks clear rules or rationale, introducing a degree of randomness to the
method’s effectiveness and making it difficult to ensure consistent performance
improvements. While random image transformations do enhance the model's adaptability,
the absence of systematic analysis on their impact across different tasks and scenarios could
undermine the method's reliability and stability. For example, the "Crop" transformation
may effectively focus on key areas in certain object detection tasks, but it could lead to
inaccuracies in tasks involving object counting or position recognition by cropping out
critical parts. Similarly, while "Horizontal Flip" can improve the model's robustness, it may
mislead the model in tasks that require maintaining the relative position or orientation of
objects.

**Questions:**

1) Don't quite understand the example presented in Figure 1. The question asks how many
green bananas are in the image, but when both the color-altered image and the original
image are input together, the result turns out to be correct, which seems counterintuitive.
2)The authenticity of the data is questionable: Table 1 in this paper mentions that LLaVA 1.5
achieved an F1 score slightly above 84 in the Random mode of the POPE benchmark, but in
the LLaVA 1.5 paper, both LLaVA-1.5-7B and LLaVA-1.5-13B scored over 87. How did this
discrepancy arise?

---

### Note · Authors · 2024-11-14

**Comment:**

We sincerely appreciate the valuable reviews from the reviewers. We will reflect the comments and suggestions to improve the manuscript.

**Withdrawal Confirmation:**

I have read and agree with the venue's withdrawal policy on behalf of myself and my co-authors.